# A theoretical model of Polycomb/Trithorax action unites stable epigenetic memory and dynamic regulation

Jeannette Reinig[1,4], Frank Ruge [2,4], Martin Howard [3] & Leonie Ringrose [1,2✉]

Polycomb and Trithorax group proteins maintain stable epigenetic memory of gene expression states for some genes, but many targets show highly dynamic regulation. Here we combine experiment and theory to examine the mechanistic basis of these different modes of regulation. We present a mathematical model comprising a Polycomb/Trithorax response element (PRE/TRE) coupled to a promoter and including *Drosophila* developmental timing. The model accurately recapitulates published studies of PRE/TRE mediated epigenetic memory of both silencing and activation. With minimal parameter changes, the same model can also recapitulate experimental data for a different PRE/TRE that allows dynamic regulation of its target gene. The model predicts that both cell cycle length and PRE/TRE identity are critical for determining whether the system gives stable memory or dynamic regulation. Our work provides a simple unifying framework for a rich repertoire of PRE/TRE functions, and thus provides insights into genome-wide Polycomb/Trithorax regulation.

[1] Humboldt Universität zu Berlin, IRI- Lifesciences, Philippstr. 13, 10115 Berlin, Germany. [2] IMBA, Institute of Molecular Biotechnology, Dr. Bohr- Gasse 3, 1030 Vienna, Austria. [3] John Innes Centre, Norwich Research Park, Norwich NR4 7UH, UK. [4]These authors contributed equally: Jeannette Reinig, Frank Ruge. ✉email: leonie.ringrose@hu-berlin.de

Epigenetic memory is essential to many biological systems, allowing maintenance of gene expression states over multiple cell generations in the absence of the initiating signals[1]. Polycomb/Trithorax response elements (PRE/TREs) are cis-regulatory elements that can maintain epigenetic memory of repressed gene expression states over many cell generations. In transgenic reporter assays using PRE/TREs from *Homeobox* (*Hox*) genes, maintenance of repression depends on the Polycomb group (PcG) proteins[2–5]. Several *Drosophila Hox* PRE/TREs have also been shown to maintain the memory of transiently activated gene expression states, in a manner dependent on the Trithorax group (TrxG) proteins[6]. Several different *Drosophila* PRE/TREs are interchangeable in these assays, suggesting that epigenetic memory is a general property of PRE/TREs[2,7–10]. The above examples from *Drosophila* have several features in common: First, the responsiveness of the system to reporter gene expression state decreases as development proceeds[3,6]. Second, the initial repressors or activators disappear during development, and expression status is maintained in their absence. Finally, the state maintained by the PRE/TRE is either on or off, and is stable over the whole of development. This has given rise to a paradigm in which PRE/TREs are thought to be switchable elements that maintain stable epigenetic memory of both silent and active states, and do so more stably as development proceeds[4,11].

However, there are also results that argue against classifying all PRE/TREs as epigenetic memory elements. Genome-wide studies of PcG and TrxG target genes in flies and vertebrates have identified several hundred targets beyond the *Hox* genes, many of which do not conform to the above criteria[12–16]. These genes include many that switch late in development, or that switch dynamically several times. This raises the question of how these genes overcome the restrictions imposed by a memory system that gains stability as development proceeds. In addition the expression patterns of many of these PcG/TrxG target genes are far more complex than a simple on or off state[14,17–20]. Finally for a large number of these genes, the transcription factors that regulate them do not disappear but are present throughout the time window of their expression or repression. Thus these genes do not appear to be subject to epigenetic memory in the classic sense, raising the intriguing question of how the PcG and TrxG proteins are involved in their regulation (reviewed in ref. [13]).

In summary, although some PcG/TrxG target genes are subject to epigenetic memory in the strict sense, many are regulated in a far more dynamic manner, suggesting a rich repertoire of PcG/TrxG regulatory modes. How a given gene responds to PcG/TrxG regulation may depend on developmental timing, transcriptional status of the associated gene, and inherent PRE/TRE properties, determined by differences in their nucleic acid sequences[4]. Understanding the mechanistic basis for these different modes of regulation will be essential for understanding genome-wide PcG/TrxG function in health and disease. A large gap in our understanding of PcG/TrxG regulation has been the lack of a coherent theoretical framework that links developmental timing and transcriptional regulation to PRE/TRE activity. We are working to bridge this gap by using simple mathematical models.

Here we present a mathematical model consisting of a PRE/TRE coupled to a promoter. The system is subjected to replication cycles whose length and number reflect those that occur during *Drosophila* development. We combine theory and experiment to quantitatively dissect the contributions of developmental timing, transcriptional input and PRE/TRE identity to the output of the system. We show that this simple model can accurately recapitulate published studies of PRE/TRE mediated epigenetic memory of both silencing and activation in *Drosophila*, and that cell cycle length is an essential component of memory. Furthermore with minimal parameter changes, the model can also precisely recapitulate our own experimental data for a different PRE/TRE that allows late switching and highly dynamic regulation of its target gene. In summary, we show that a single simple model can account for profoundly different regulatory modes, and we identify parameters that govern those differences. Thus, this work has broad implications for understanding the molecular nature of locus-specific and developmental differences in stability and flexibility of genome-wide PcG/TrxG regulation.

## Results

**A model for PcG/TrxG regulation links promoter and PRE/TRE**. To examine the regulatory interactions between PRE/TREs and their target promoters during *Drosophila* development, we used minimal stochastic models for a promoter and a PRE/TRE, and introduced coupling between them. The model, its implementation and the assumptions used are described in detail in Supplementary Methods. Here we summarise the most important features.

The promoter was modelled as an array of DNA binding sites for a transcription factor (Fig. 1a), each of which can be either free (F) or bound (B). The probabilities of binding and unbinding are represented by the parameters $p1$ and $p2$, respectively, which can be modified to reflect different promoter strengths (Fig. 1b). The promoter state (active or repressed) is given by the proportion of sites that are in the B or F configurations, respectively.

The PRE/TRE was modelled as an array of nucleosomes, as described in refs. [21,22] (Fig. 1a). Despite its complexity, the PcG/TrxG system is robustly bistable[23], thus a simple bistable model is reasonable. Each nucleosome can be in a silent (M), neutral (U) or active (A) configuration. The PRE/TRE state (active or silent) is given by the proportion of nucleosomes that are in the A or M configurations, respectively. Each of the A and M configurations represents all histone modifications and other bound molecules, such as TrxG and PcG proteins or non-coding RNAs that contribute to activation or silencing. Thus the model makes no assumptions about the molecular nature of memory. We use whole nucleosomes as the minimum unit, as in refs. [21,22] (see Supplementary Methods for more detail). In the model, a nucleosome in the M or A configuration will attempt to convert other nucleosomes in the array towards its own configuration, with probabilities $p3$ and $p4$ for M and A respectively (Fig. 1b). This feedback renders intermediate U nucleosome unstable, and A and M nucleosome configurations much more stable[21,22]. Thus the PRE/TRE tends to adopt a dominant A or M state, and is bistable. A further parameter ($p5$) gives the probability of interconversions between A, U and M configurations that are independent of feedback (Fig. 1b). This includes histone exchange and random noisy conversions as described previously[21,22] but also includes specific conversions that do not require a previously existing modification for a modifying enzyme to be recruited, such as direct recruitment by DNA binding proteins[24,25].

The separation of promoter and PRE/TRE in the model reflects the regulatory units that are experimentally tractable, and allows the effect of regulated coupling between them to be examined in simulations. The PRE/TRE and promoter were coupled to each other as described in Supplementary Methods, Fig. 1c and Supplementary Fig. 1. The effect of coupling is to render the promoter and the PRE/TRE dependent on each other, so that the more active or silent the PRE/TRE, the more active or silent the promoter, and *vice versa*. This reflects the regulatory interactions that have been observed in vivo[2,3,6]. The exact mechanism of this coupling is not known but may include looping, spreading of chromatin marks or interaction of homologs. These mechanisms are not explicitly included in the

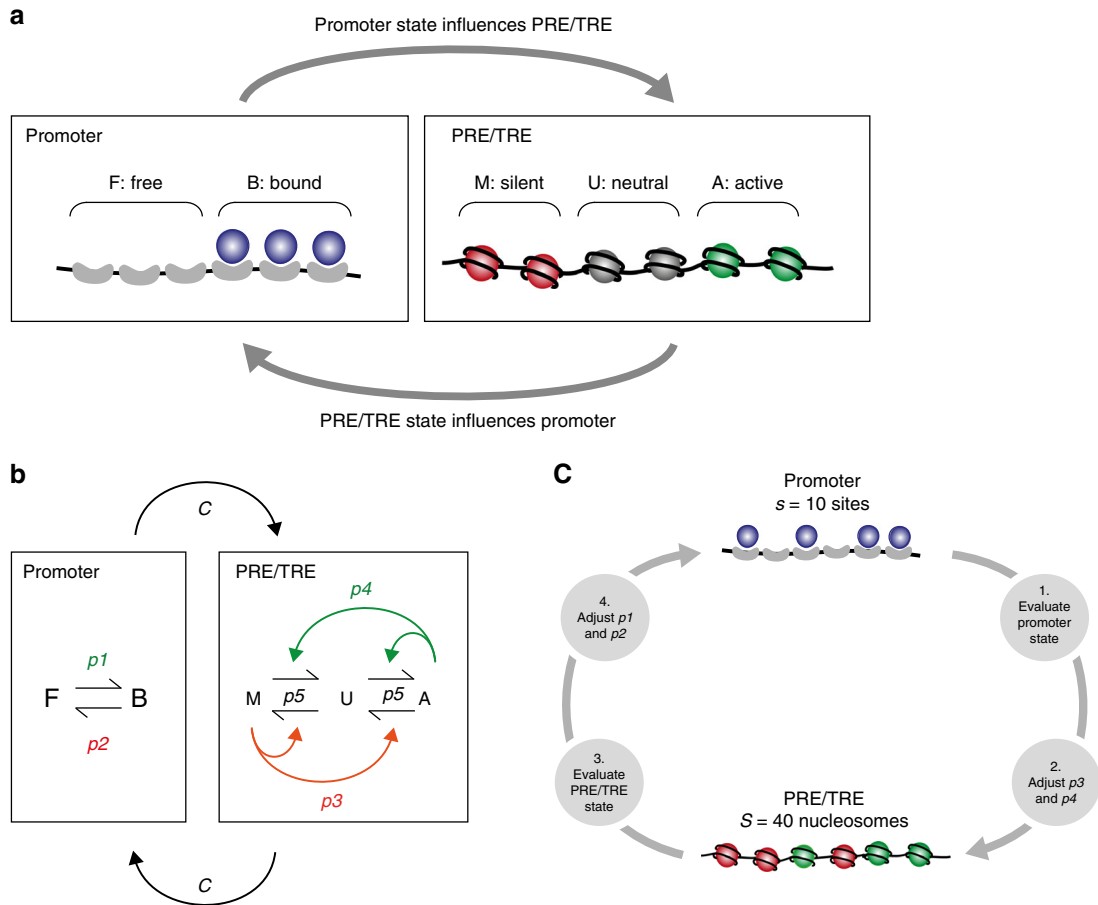

**Fig. 1 A simple model for Polycomb/Trithorax regulation. a** The promoter and PRE/TRE are shown schematically. Left: each promoter site can be either free (F) or bound (B). Right: each nucleosome in the PRE/TRE can be either silent (red, M), neutral (grey, U) or active (green, A). See main text for details. **b** The model is implemented stochastically, with probabilities for each of the transitions between F and B, and between A, U and M. For the promoter, *p1*: probability of transcription factor binding at a single promoter binding site. *p2*: probability of transcription factor unbinding at a single site. For the PRE/TRE, *p3* and *p4* (red and green arrows) denote feedback reactions in which nucleosomes in each of the M or A configurations convert other nucleosomes towards that configuration. The parameter *p5* (black arrows) gives the probability of conversions between A, U and M that are independent of feedback. *C*: The promoter and PRE/TRE are coupled. See also Supplementary Fig. 1. **c** At each iteration of the simulation, the promoter state is evaluated and used to adjust the PRE/TRE parameters *p3* and *p4*. Likewise the PRE/TRE state is evaluated and used to adjust the promoter parameters *p1* and *p2*. These adjusted *p1, p2, p3* and *p4* values are used in the next iteration. For coupling relationships see Supplementary Fig. 1.

model, but the mathematical description of the interaction between PRE/TRE state and promoter, and the strength of this interaction, were varied to identify the model that best fits the data. 14 different coupling models were explored (Supplementary Fig. 1), which differ in the number of model parameters (*p1, p2* and (*p3, p4*)) that are adjusted at each iteration (Supplementary Fig. 1G, H), and the mathematical relationship between PRE/TRE state and promoter state (Supplementary Fig. 1A, B). This analysis showed that only two mathematical descriptions of coupling gave robust results in all tests, designated as models 1 and 2 in the rest of this paper. Models 1 and 2 have in common that they adjust all of the four model parameters at each iteration (*p1, p2* and (*p3, p4*)). Model 1 is used for all results shown in the main figures, models 1 and 2 are compared in Supplementary Figs. 3 and 8. Coupling strength in the model is adjusted by the parameter *C* (Fig. 1b), which determines the magnitude of the response of the promoter to a given PRE/TRE state, and that of the response of the PRE/TRE to a given promoter state.

To take account of the changes in cell cycle length during *Drosophila* development, we curated published data on cell cycle timing for all developmental stages as described in detail in Supplementary Methods. These time constraints for each cell

cycle were included in the model (Supplementary Table 1). To model cell divisions for the promoter, all sites were set to F (unbound) at the end of each cell cycle. This is based on the observation that many transcription factors dissociate from mitotic chromatin[26] and that transcription is actively and globally repressed during mitosis[27,28]. For the PRE/TRE, cell division was modelled as described previously in refs. [21,22], by setting each nucleosome to U with a probability of 0.5, at the end of each cell cycle. Thus on average, half of the nucleosomes are set to U at the end of each cell cycle. This is based on the observation that parental histones and their modifications are partitioned randomly to the two daughter chromosomes after replication[4,29]. In summary, this simple model comprises several essential features of PRE/TRE mediated gene regulation during *Drosophila* development, namely a regulatable promoter coupled to a bistable PRE/TRE, and the known timing of cell cycles throughout development.

**The model recapitulates memory of silencing**. We first asked whether the model can recapitulate the epigenetic memory of silencing shown by a PRE/TRE in a transgenic *Drosophila*

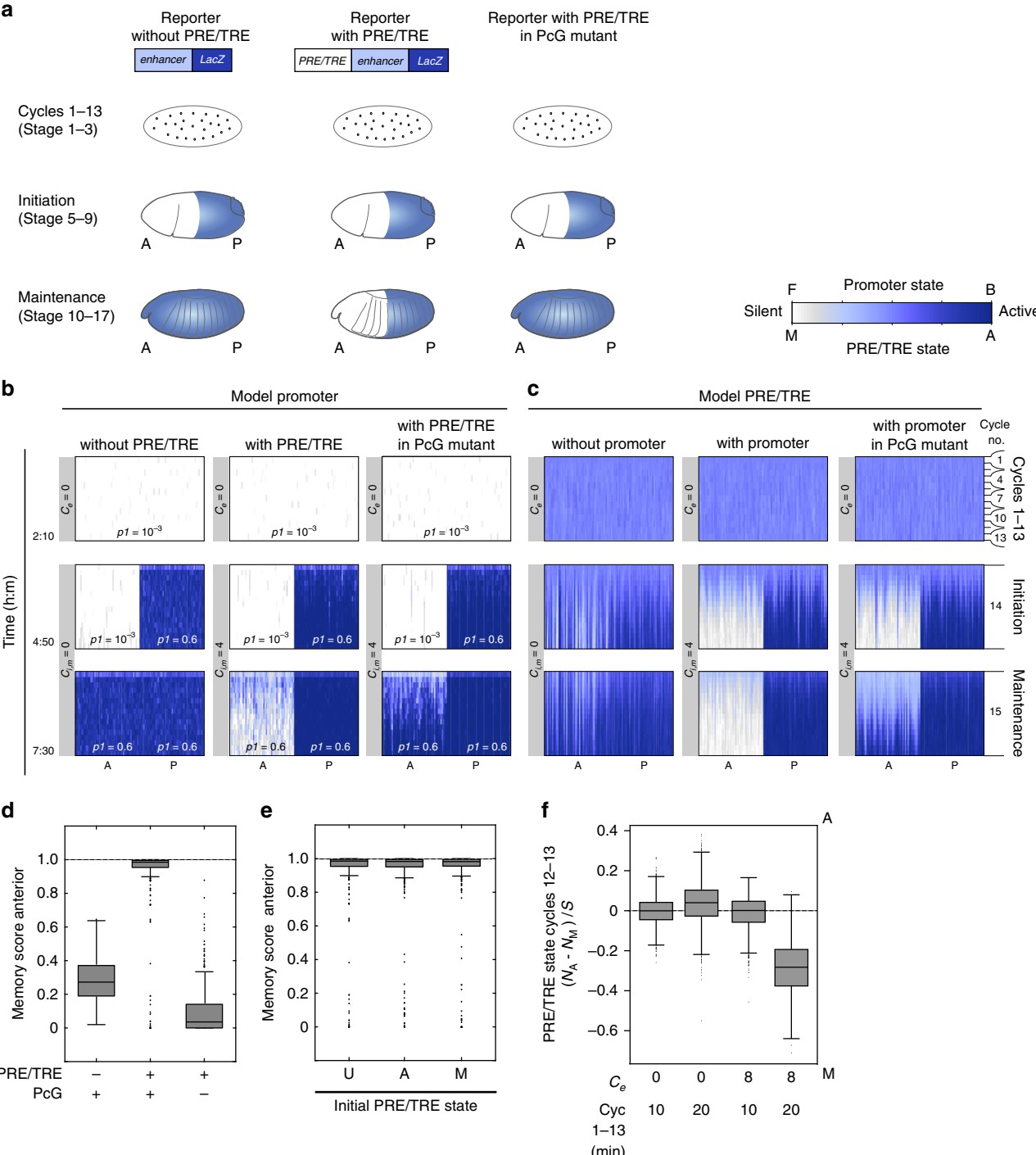

**Fig. 2 The model recapitulates memory of silencing. a** Experimental test of epigenetic memory of silencing[2,3]. See main text for details. Top: transgenic *LacZ* reporter constructs. Bottom: embryonic *LacZ* patterns. Active *LacZ*: blue. Anterior: A, posterior: P. **b, c** Simulated time courses of *Drosophila* development showing promoter state (**b**) and PRE/TRE state (**c**) over time. Active: blue; silent: white. Division cycles are indicated (right). Plots show results from model 1 (see also Supplementary Figs. 1 and 2). Initial conditions, promoter: all sites are F; PRE/TRE: all nucleosomes are U. **b** Promoter state. The input values of $p1$, $C_e$ and $C_{i,m}$ are shown on the plots, $p2$ (TF dissociation) was 0.1 in all simulations (see also Supplementary Table 2). Input values of $p3$ and $p4 = 0.25$, $p5 = 0.04$. Simulations were performed without coupling between promoter and PRE/TRE (left), and with coupling during the initiation and maintenance phases (middle). Right: PcG mutant was simulated by reducing $p3$ to 0.001 at the onset of the maintenance phase. (**c**) PRE/TRE state. Parameter values as in (**b**); (see also Supplementary Table 2 and Supplementary Fig. 2). For each condition, data from 50 independent simulations are shown. Promoter and PRE/TRE data from the same simulation run are shown. Each of the 50 simulations is continuous throughout the vertical scale. **d, e** Memory score in anterior compartment (1 indicates perfect memory, see Supplementary Fig. 2 for calculation). Boxplots of data for 400 independent simulations. Central mark on box plots: median; bottom and top edges of box: 25th and 75th percentiles, respectively. Whiskers extend to cover 99.3% of the data. Outliers are plotted as dots. **f** PRE/TRE states for the different early cycle conditions as indicated, averaged over cycles 12 and 13 for 1000 independent simulations. PRE/TRE state (on a scale of −1 to +1, where −1 is silent, and +1 is active). $N_A$ = number of A nucleosomes, $N_M$ = number of M nucleosomes, $S$ = total number of nucleosomes. Boxplot parameters as in (**d, e**).

reporter assay[2,3] (Fig. 2). In the experiment, an embryonic enhancer from a *Hox* gene (such as *Ultrabithorax (Ubx))* is linked to a *β-galactosidase (LacZ)* reporter gene, with or without a PRE/TRE (Fig. 2a). Transcription is generally silent during the first 13 cycles before zygotic genome activation at cycle 14[30]. During the initiation phase, the *Ubx* enhancer is activated in the posterior part of the embryo and suppressed by repressors in the anterior (Fig. 2a, left and Supplementary Table 1). At the onset of stage 10, the repressors disappear and the reporter becomes active in the anterior unless coupled to a PRE/TRE (Fig. 2a, middle and Supplementary Table 1). The maintenance of silencing is dependent on the PRE/TRE, and is lost in a PcG mutant (Fig. 2a, right)[2,3].

To adapt the model to the experimental observations, we used the model promoter to represent the *LacZ* reporter gene described above. We first established the pattern of the model promoter throughout development without the PRE/TRE by adjusting the input value of the parameter *p1*, representing promoter repression (small *p1*) or activation (large *p1*) (Fig. 2b, left panels and Supplementary Table 2). To ask whether the model PRE/TRE is able to maintain memory of anterior repression of the promoter that was established during the initiation phase, we coupled the PRE/TRE to the promoter and ran the simulation with the same promoter inputs, and various coupling regimes and PRE/TRE parameters (Fig. 2b, middle and Supplementary Fig. 2).

To reduce the number of free parameters for this analysis, we introduced several constraints (see Supplementary Fig. 2 and Supplementary Methods). The input values of all PRE/TRE parameters were kept constant throughout development, and the feedback parameters *p3* and *p4* (for silencing and activation respectively) were kept equal to each other. The coupling strength *C*, of PRE/TRE to promoter and promoter to PRE/TRE was kept equal in both directions, and this strength was kept constant during the initiation and maintenance phases (Supplementary Fig. 2). Coupling during the first 13 cycles was set to 0 (this requirement was determined by fitting, see Fig. 2). Thus the model was fitted with only three free parameters: (*p3*, *p4*) (feedback); *p5* (feedback-independent transition) and $C_{i,m}$ (coupling during initiation and maintenance phases). An additional constraint in the parameter search was the requirement for a uniform and spatially precise memory of the initiated pattern until the end of the maintenance phase under the influence of the PRE/TRE (Fig. 2b and Supplementary Fig. 2).

Under these conditions, a range of values for the parameters (*p3*, *p4*), *p5*, and $C_{i,m}$ were found, under which the system gave a precise memory of the promoter expression pattern established during the initiation phase, namely repression in the anterior and uniform high expression in the posterior (Supplementary Fig. 2 and Fig. 2b, middle panels). This maintenance was stable until 7h30 of development in the model (Fig. 2b, middle panels) and throughout adult development despite several further replication cycles (for examples of persistence of early established states until later development see Fig. 3d). Reduction of the parameter *p3* to simulate PcG null mutants that display loss of repression at the onset of the maintenance phase led to a derepression of the model promoter in the anterior compartment after ~6 h of the simulation, consistent with published data (Fig. 2b, d and Supplementary Table 2)[2,3].

The system gave optimum memory over a range of values for (*p3*, *p4*) and *p5*. However, these parameters also showed upper and lower limits, due to the stringent criteria that must be fulfilled for memory to be effective (Supplementary Fig. 2). The feedback-dependent parameters *p3* and *p4* determine the speed with which the PRE/TRE can be converted to one state or the other. This is critical in the model, as the PRE/TRE has to be fully converted to

a stable A or M state within the time frame of the initiation phase (cycle 14, Fig. 2c, middle). However, if this conversion rate is too fast, then random A and M states that exist at the end of cycle 13 become fixed before the PRE/TRE can receive information from the promoter, causing variegation. The parameter *p5* (feedback-independent transitions) affects the stability of the system. If *p5* is low, then feedback dominates, and the system has a high capacity for memory but is unable to respond to changes in the promoter state during early initiation (cycle 14). If *p5* is high, the opposite is true: the PRE/TRE responds well to the promoter but is unable to maintain memory (Supplementary Fig. 2). A minimum value of $C_{i,m}$ (coupling strength during initiation and maintenance phases) was required for optimum memory (see Supplementary Fig. 2D). At lower $C_{i,m}$-values the system responded too slowly to changes in promoter or PRE/TRE state, and memory was not correctly established during the initiation phase. Taken together these results demonstrate that the simple model is sufficient to recapitulate several essential features of epigenetic memory of silencing that have been observed experimentally, and define a set of conditions under which memory is most effective.

**Early rapid division cycles are essential flexibility.** In the above simulations, the early rapid cycles 1–13 and absence of coupling during these cycles had the effect of keeping the PRE/TRE predominantly in the U configuration prior to the initiation phase (on average approximately 90% of nucleosomes were in the U (unmodified) state at any given time, Fig. 2c). To gain further insight into these early stages, we varied the strength of coupling and the length of the early cycles in simulations.

This analysis revealed that longer initial cycles allowed the PRE/TRE to stabilise more often into random A or M states before being coupled to the promoter, causing variegation (because the PRE/TRE had more time to stabilise before being wiped by replication; Fig. 2f). Introducing coupling during these early cycles, even without a change in cycle length, resulted in a bias towards M states, resulting in a failure to adopt the A state in response to the active promoter in the posterior (because the PRE/TRE was coupled early to a strongly silenced promoter). This effect became more pronounced if cycle length was increased in addition to coupling (Fig. 2f).

In summary this analysis demonstrates that in the model, the early rapid division cycles and an absence of coupling during this phase are essential for keeping the PRE/TRE in a naive state prior to receiving information from the promoter, and are thus instrumental in determining system flexibility and fidelity of epigenetic memory.

**The same model recapitulates memory of activation.** We next asked whether the model can recapitulate the epigenetic memory of activation shown by a PRE/TRE in a transgenic *Drosophila* reporter assay[6], and if so, with which parameters. In the activation assay, the yeast GAL4 transcription activator protein is expressed under control of a heat shock promoter (Fig. 3a). The reporter construct carries an upstream activating sequence (UAS), which is activated by GAL4, upstream of a PRE/TRE and two reporter genes. The *LacZ* reporter enables detection of expression in embryos and larvae, and the *miniwhite* (*mw*) reporter gives a red pigment in adult eyes (Fig. 3b). In the absence of heat shock, both reporters are silenced by the PRE/TRE, and the embryo and the adult eyes are mostly light in colour with some variegation (Fig. 3b, left). If a 1-h heat shock is given during late developmental stages, e.g., in the larva, then the reporter is transiently activated but is re-silenced before the adult stage (Fig. 3b, middle). In contrast, if a 1-h heat shock is given during embryonic stages then the reporter is activated and remains active

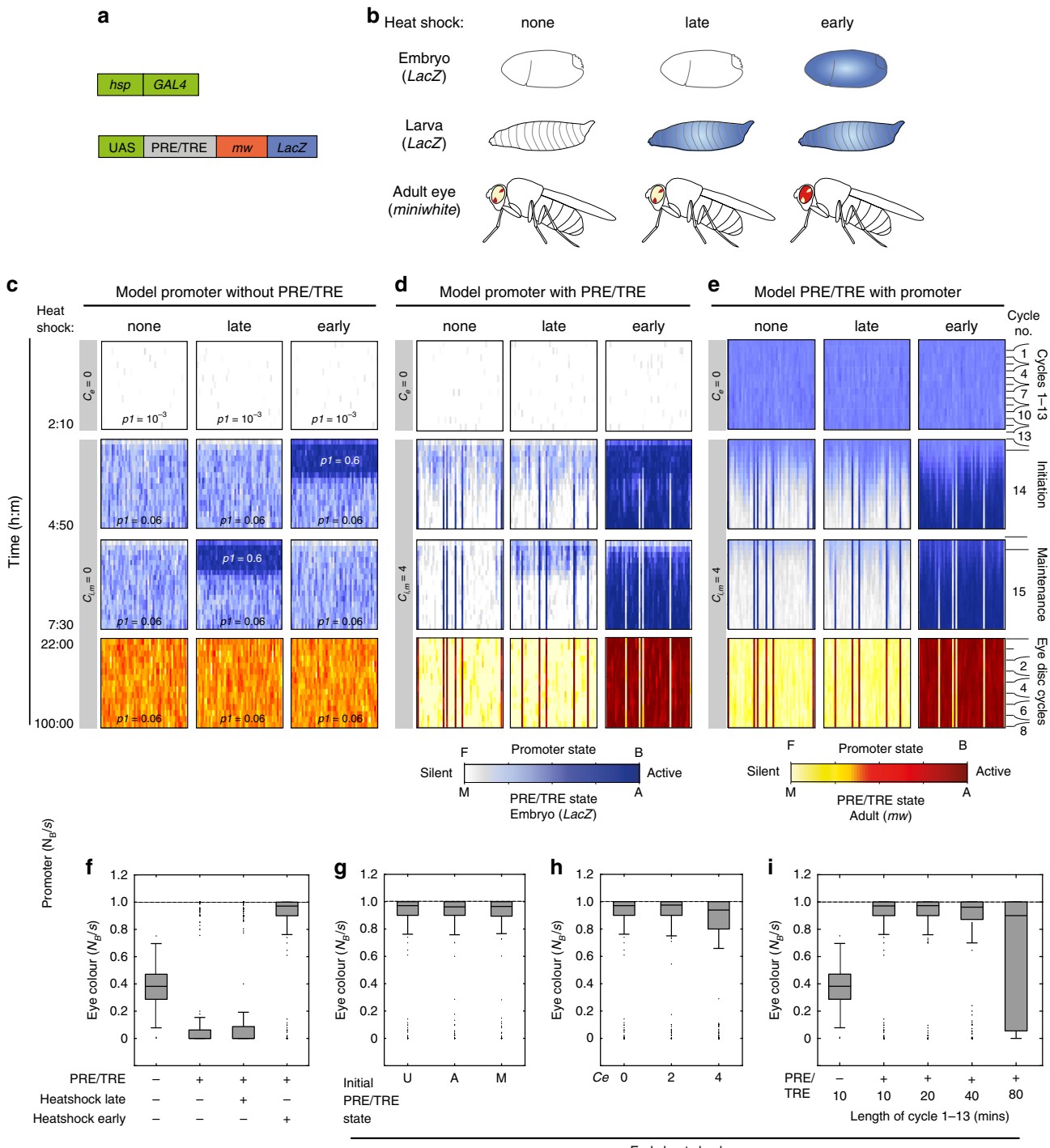

**Fig. 3 The same model recapitulates memory of activation. a**, **b** Experimental test of epigenetic memory of activation[6] (see main text for details). **a** Heat-shock GA-4 and UAS PRE/TRE reporter constructs. *Hsp*: heat shock promoter. *mw*: miniwhite. **b** Summary of results in ref. [6]. Active *LacZ*: blue; active *mw*: red. **c**–**e** Simulated time courses of *Drosophila* development showing model promoter activity (**c**, **d**) and PRE/TRE activity (**e**) over time for the first 7.5 h of development and for the eye disc up to mid third instar (22–100 h). Division cycles are indicated (right). Plots show results from model 1 (see also Supplementary Figs. 1 and 2). The simulation was run continuously from 0 to 100 h, but only the time windows indicated are shown. 22 h–100 h: dark red: active; yellow: silenced, to represent *miniwhite* activity. For each condition, data from 50 independent simulations are shown, each of which is continuous throughout the vertical scale. The input value of the parameter *p1* was varied as indicated on the plots in (**c**). The same *p1* values were used for the corresponding time segments in (**d**, **e**). All other parameter values and initial conditions were as in Fig. 2b, c (middle). $C_e$ and $C_{i,m}$ values are indicated. **f**–**i** Boxplots of promoter state averaged over 90–100 h for 400 independent simulations. In simulations, the promoter level at end of third instar (100 h) did not change over further developmental time, thus it is used here as a proxy for adult eye colour. $N_B$: number of promoter sites in B configuration, *s*: total number of sites. Boxplot parameters as in Fig. 2d, e. **f** Effect of heatshock timing. **g**–**i** Under conditions in which a heatshock was given early, and the PRE/TRE was coupled to the promoter at the onset of initiation phase, different additional parameters were varied. **g** Initial state of all nucleosomes in the PRE/TRE set to A, U or M. **h** $C_e$, was varied as shown. Later coupling was not varied ($C_{i,m} = 4$). **i** Length of cycles 1–13 was varied as indicated. See also Supplementary Figs. 1 and 3.

throughout development, giving a proportion of adults with red eyes and some variegation (Fig. 3b, right)[6]. Several different PRE/TREs have been shown to give memory of activation in this assay[6,10,17]. The effectiveness of heat shock during embryonic but not larval stages has been proposed to indicate a qualitative change in the mechanism of PRE/TRE action during development[6].

To ask whether the model can recapitulate these observations we adapted the model promoter without the PRE/TRE to the experimental heat shock conditions (without heatshock, with late or with early heatshock; Fig. 3c). We then evaluated the effect of coupling a PRE/TRE to this model promoter in time courses simulated for the whole of development, including the adult eye (Fig. 3c-e and Supplementary Tables 1 and 2, Supplementary Methods). Remarkably, the same values of the four parameters ($p3$, $p4$) (feedback) $p5$ (feedback independent transitions) $C_e$ (early coupling) and $C_{im}$ (late coupling) as used above (Fig. 2b, c (middle)) were also able to accurately recapitulate the experimentally observed memory of heat shock-induced activated states (Fig. 3d). In particular, a 1-h heat shock given after the end of the initiation phase (4h50) allowed transient activation of the reporter but failed to induce stable switching and memory (Fig. 3d, late). In contrast, a 1-h heat shock given during the initiation phase (2h10 to 4h50) gave stable activation and memory, even after the removal of the heat shock stimulus (Fig. 3d, early). Parameter space analysis revealed a similar optimum range of values for the three free parameters as those defined for memory of silencing (Supplementary Fig. 3).

Similarly to the silencing experiment described above, the strength of memory was independent of initial PRE/TRE state (Fig. 3g), but was dependent on coupling and length of the first 13 cycles (Fig. 3h, i). We conclude that the same model, with the same parameter values and identical cell cycle constraints are sufficient to recapitulate memory of both silencing and activation.

**Chromatin modifications increase during development.** Several previous theoretical studies of chromatin-based epigenetic memory have shown that stability of memory in bistable model systems increases with increasing cell cycle length[21,31]. However, these models have not been applied to the observed real changes in cell cycle length that occur during development. To examine the effect of developmental changes in *Drosophila* division cycle length on the rate of global accumulation of A and M states in the model, we ran simulations over a developmental time course. We ran 1000 simulations on a nucleosomal array that was not coupled to a promoter, over 11 h of development and scored the average levels of A and M states at different time points (Fig. 4a). This showed that in the model, A and M states are present at low levels from the onset of development and accumulate slowly during development, reaching maximum levels only during late embryonic development. This is consistent with the observation that heterochromatin features (HP1 and H3K9 methylation) accumulate slowly during cycles 11–13 and increase substantially during cycle 14[32,33], and with ChIP[34] and mass spectrometry analysis[35] showing that several other histone modifications also accumulate during *Drosophila* development. The model further predicts that histone modifications will accumulate earlier in pole cells, which exit the cell cycle at cycle 10 and commit to the germ cell fate (Fig. 4a)[36].

To test the model predictions in individual animals, we analysed the accumulation of the Polycomb (PC) protein and of histone modifications in stage 2–14 *Drosophila* embryos by immunofluorescence (Fig. 4 and Supplementary Figs. 4–6). We analysed the PC protein and histone modifications associated with silencing (H3K27me3), and activation (H3K36me2, H3K36me3,

H3K4me1 and H3K4me3). Unmodified histone H3 served as an internal control. The results are summarised in Fig. 4 and Supplementary Fig. 5.

Neither histone modifications nor the PC protein showed any detectable signal in interphase nuclei during cycles 1–13 (Fig. 4b, c and Supplementary Fig. 5). All accumulated with different kinetics over the subsequent developmental stages. At 11 h (late maintenance phase), all nuclei contained visible signals for all marks. Interestingly, the earliest detectable signal for several modifications was in the pole cells, typically becoming visible at stage 5–6 (early cycle 14), before the somatic nuclei showed a signal (Fig. 4b-d and Supplementary Fig. 5). This observation is consistent with the prediction of the model (Fig. 4a), and suggests that early cell cycle exit may contribute to accumulation of chromatin modifications.

In somatic cells, whilst PC, H3K4me1 and H3K4me3 were detectable in interphase nuclei at stages 5–6, the other modifications (H3K36me and H3K27me3) first became detectable in a subset of interphase nuclei at stage 9 (Fig. 4d). The same modifications became visible earlier at stages 6–7 in mitotic nuclei (ref. [37], marked 'm' in Fig. 4 and Supplementary Fig. 5). The appearance in mitotic nuclei before detection in interphase is likely due to the condensation of mitotic chromatin, making the epitope more readily detectable. We note that[38] also detected H3K27me3 in early mitotic nuclei and in pole cell nuclei, however, immunofluorescence data on accumulation in interphase nuclei were not presented, and time points beyond stage 5 were not analysed in that study.

We did not detect histone modifications by immunofluorescence during cycles 1–13. Several authors have observed histone modifications present in chromatin during these early cycles by ChIP[34,38]. However, these studies did not address time points later than stage 5 (2h50, see Supplementary Fig. 6), and a recent study performing mass spectrometry did not address the early cycles 1–3[35]. To evaluate PC and histone modifications across a wider time window, we performed ChIP followed by whole-genome qPCR on staged embryo collections (Fig. 4e, see "Methods"). This analysis showed that PC and all histone modifications were detectable on chromatin during the first 2 h of development (stages 1–4, corresponding to cycles 1–13), and subsequently accumulated at different rates, reaching maximal levels at 8–10 h. The H3K4me1 and me3 modifications were detectable at earlier stages than H3K27me and H3K36me, consistent with our immunofluorescence data and with a recent ChIP study using more highly resolved time windows[34] (Supplementary Fig. 6). We note that different histone modifications and PC show different kinetics of detectable accumulation in both the immunofluorescence and ChIP assays, whereas the model predicts a smooth increase of all states during cycles 5–9. However, in the model, the 'A' and 'M' system states comprise all modifications and bound proteins that contribute to that state, and thus in its current form the model does not enable predictions about separate molecular events. The important similarities of our own and other published data to the model prediction are that the PC protein and specific histone modifications are present on chromatin at low levels during the early cycles 1–13, accumulate during development, and reach maximum levels during the maintenance phase.

**Two PRE/TREs respond differently to dynamic input.** We have established model conditions that recapitulate two classical paradigms of PRE/TRE mediated regulation, namely epigenetic memory of silent and of active promoter states. To evaluate whether the model is applicable to a more dynamic mode of regulation we challenged it with an extreme case: the *eyes absent*

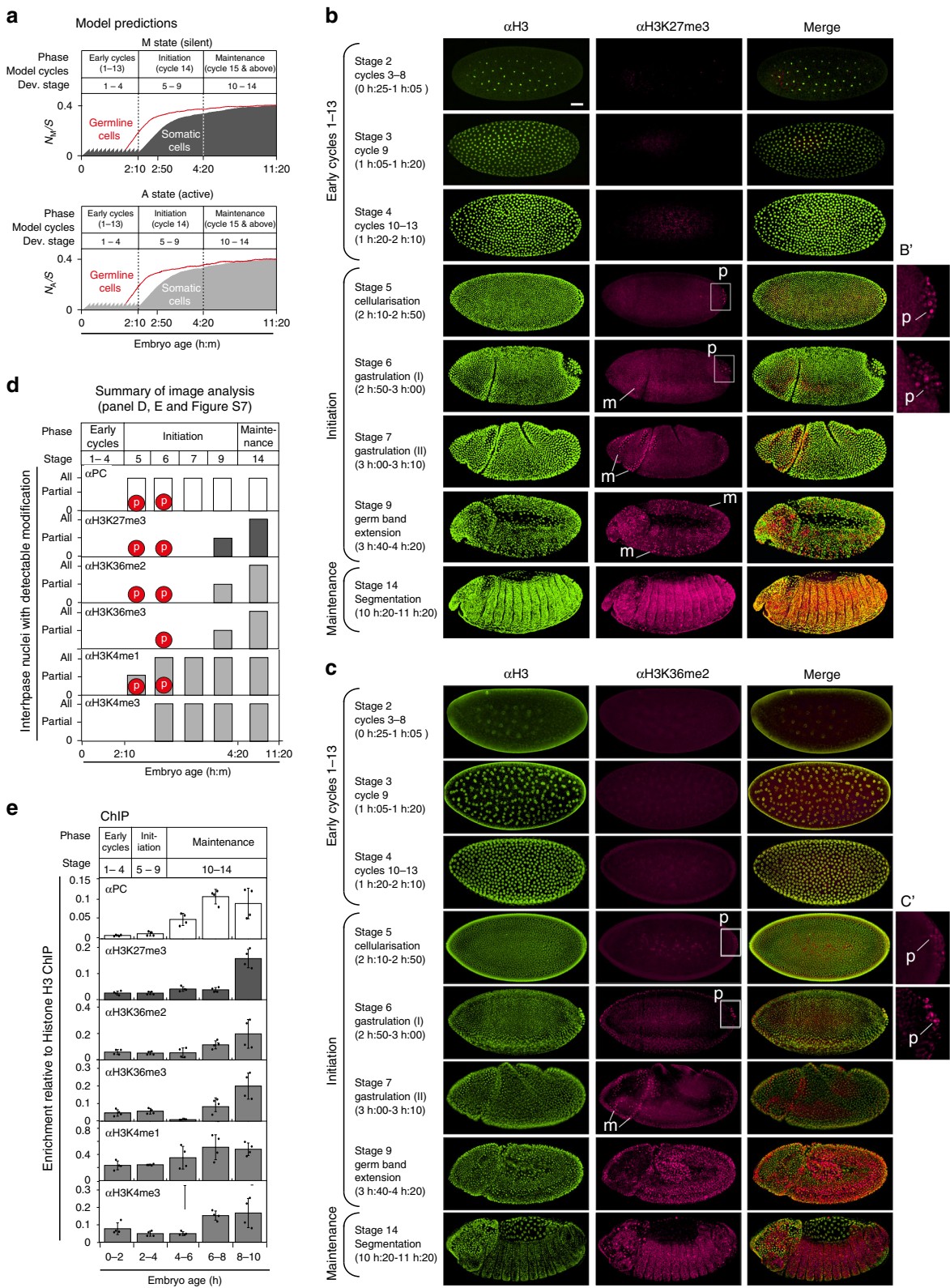

(eya) gene (Fig. 5). The eya gene is fundamentally different in its regulation to the Hox genes: it switches in specific cells from a silent to an active state late in development[39], it does not appear to require a classical memory of activation and repression because its activators and repressors remain present[13,40], and it displays a gradient rather than an all-or-none pattern[39], Fig. 5b (for more details on eya regulation see Supplementary Methods).

The eya 5′ region contains two potential PRE/TREs that are enriched for PcG and/or TrxG proteins in ChIP-seq data sets (Fig. 5a). One putative PRE/TRE is at the promoter and a second is in the first intron. The intronic PRE/TRE is well-characterised in reporter assays and contains a high density of binding sites for the PHO, GAF and Zeste proteins[12,15] (Fig. 5a). We chose to experimentally analyse the function of the intronic PRE/TRE in

**Fig. 4 Chromatin modifications increase during *Drosophila* development. a** Simulated time course of accumulation of M (top) and A (bottom) nucleosomes as a proportion of total nucleosomes (*S*) in the array over developmental time, averaged over 1000 individual simulations. Solid grey: somatic cells, red line: germline cells, modelled by implementing early cycles 1–9, followed by cycle 10 of 2 h, and subsequent G2 arrest for the rest of the simulation[36]. Parameters as in Fig. 2b, c (no coupling between PRE/TRE and promoter). **b, c** Embryos were fixed and double stained with antibodies against unmodified histone H3 (green, left panels) and H3K27me3 (**b**) or H3K36me2 (**c**) (magenta, middle panels). Merge: right panels. Embryos were staged according to morphology, as indicated. Three slides per antibody were prepared, typically containing 50–100 embryos, of which 5–10 were at the required stage and all showed similar staining for a given stage. White boxes at stages 5 and 6 indicate pole cells (p), which first become apparent at stage 5. B′ and C′ show 3× zoom of boxed area. Mitotic cells are indicated at stages 6–9 where visible (m). Scale bar represents 75 μm and is the same for all panels except B′ and C′. **d** Summary of immunofluorescence analysis shown in (**b, c**) and Supplementary Fig. 5. Red circles, p, indicate the stage at which staining of each modification was visible in pole cell nuclei. Stages at which histone modifications became visible in interphase nuclei are indicated (white or grey bars). "Partial" indicates that a proportion of nuclei showed detectable histone modifications. "All" indicates that all nuclei contained signal for the modification or protein. **e** ChIP analysis of PC and histone modifications in staged embryos as indicated. Global levels are shown as a proportion of histone H3 ChIP for each stage (see "Methods"). Data are presented as mean values ± SD of two IPs each made from two independent chromatin preparations (= four independent IPs in total). Individual data values are shown as black circles.

---

this study, as the putative promoter PRE/TRE spans the transcription start site, which is required for reporter expression, and thus its deletion would disrupt reporter gene transcription. An upstream enhancer drives *eya* expression in the developing larval eye disc[39] (Figs. 5a, and 6a–c).

Before implementing the model, we first dissected the contributions of the enhancer and the intronic PRE/TRE to the *eya* expression pattern experimentally. To this end, we placed a *green fluorescent protein (GFP)* reporter gene under control of *eya* regulatory sequences by generating the *eya1::GFP* reporter construct. The reporter contains 9.2 kb of genomic *eya* sequence, including the *eya* enhancer, promoter, putative TSS PRE/TRE, exon 1 and the intronic PRE/TRE. A *TurboGFP* reporter gene is fused in frame to the first three codons of *eya* exon 2, and a second *mw* reporter enables analysis in adults (Figs. 5c and 6c). To exclude genomic position effects, transgenic flies were generated in which the *eya1::GFP* construct and variants lacking the PRE/TRE or the enhancer were integrated into the same genomic location as described in[41] (Figs. 5c–f and 6c). The *eya1::GFP* reporter construct expressed GFP in a similar pattern to wild type *eya* mRNA, in a manner dependent on the presence of the *eya* enhancer (Fig. 5b–j).

Deletion and mutation analysis of the *eya1::GFP* transgene confirmed that the intronic PRE/TRE is required for PcG-dependent repression of the *mw* reporter. In the absence of this intronic PRE/TRE (Fig. 5d, l), or in a version in which binding sites for the PcG protein PHO were mutated, the *mw* reporter was strongly derepressed (Fig. 5s, t and Supplementary Fig. 7D). To evaluate genetic interactions, we introduced each variant of the *eya1::GFP* transgene into a heterozygous mutant background for the PcG gene *polymoeotic (ph410)*. Constructs containing the enhancer were highly expressed and showed no further activation in a *ph410* mutant background (Fig. 5o, p). The construct lacking both the enhancer and the intronic PRE/TRE was expressed at lower levels but also showed no response to the *ph410* mutation (Fig. 5r). In contrast, the construct lacking the enhancer but containing the intronic PRE/TRE showed derepression in a *ph410* mutant background (Fig. 5q). Taken together these results indicate that the intronic PRE/TRE plays a role in PcG mediated regulation of the *eya1::GFP* reporter.

To obtain quantitative information, we performed in situ hybridisations against the *GFP* mRNA and compared the patterns with and without the intronic PRE/TRE. Interestingly, the PRE/TRE had the effect of sharpening the gradient pattern established by the enhancer: the level of activation ahead of the morphogenetic furrow was higher, and the expression level posterior to the furrow was lower than in the construct without the PRE/TRE (Fig. 6d–f). We conclude that the intronic PRE/TRE fine-tunes the expression pattern of *GFP* in the *eya1::GFP* reporter construct.

To compare the properties of different PRE/TREs in this assay, we replaced the 1.5 kb *eya* intronic PRE/TRE in the reporter construct with a 1.5 kb fragment of the *bithoraxoid (bxd)* PRE/TRE, which shows memory of both silencing and activation in the assays described in Figs. 2 and 3[2,3,10]. Surprisingly, the *bxd* PRE/TRE caused strong variegation of *GFP* expression (Fig. 6g, h). Interestingly, this variegation was also graded across the eye disc, with cells that express the reporter gene ahead of the morphogenetic furrow doing so at a higher level than those behind it (Fig. 6g). We conclude that the *bxd* PRE/TRE acts as a bistable element in this assay, but is nevertheless receptive to gene expression level differences across the gradient. Thus the two PRE/TREs we have tested behave very differently when coupled to the dynamically regulated *eya* promoter and enhancer.

**The same model recapitulates the behaviour of both PRE/TREs.** We next asked whether our simple model can capture these differences, and if so, with which parameters? We implemented two versions of the model, one in which the model reporter locus contains two PRE/TREs (one at the TSS and one intronic, Fig. 5a), and a second in which it contains only one intronic PRE/TRE. Both models reached similar conclusions regarding the properties of the *eya* and *bxd* PRE/TREs (Supplementary Fig. 10 and Supplementary Methods). Thus we describe the results of the simpler, one-PRE/TRE model here. The two-PRE/TRE model is described in detail in Supplementary Methods.

We first adapted the model to recapitulate the events involved in eye disc proliferation and differentiation (see Supplementary Methods). We then modelled the observed gradient shape of the *eya1::GFP* reporter gene without the intronic PRE/TRE in the 3rd instar larval eye disc. To this end, we dynamically adjusted the parameter *p1* to create a gradient across the simulated eye disc, and fitted this gradient to the experimentally observed shape (Fig. 6 and Supplementary Fig. 8, Supplementary Methods). To examine the effect of the model PRE/TRE on this gradient, we coupled the PRE/TRE to the promoter and ran simulations for the whole of development from embryonic cycle 1 up to mid 3rd instar. We searched for PRE/TRE parameter combinations that would recapitulate the experimentally observed effect of the *eya* PRE/TRE on the promoter, namely to sharpen the anterior-posterior gradient in a smooth manner (Fig. 6f and Supplementary Fig. 8).

This analysis showed that the model can indeed recapitulate the smooth and sharpened gradient with specific parameters (Fig. 6m, n and Supplementary Fig. 9). Interestingly the optimum parameter range differed from that defined previously for the memory assays (Figs. 2 and 3) in several respects. First, the timing and strength of coupling between the PRE/TRE and the promoter was critical. The best results were obtained when coupling was

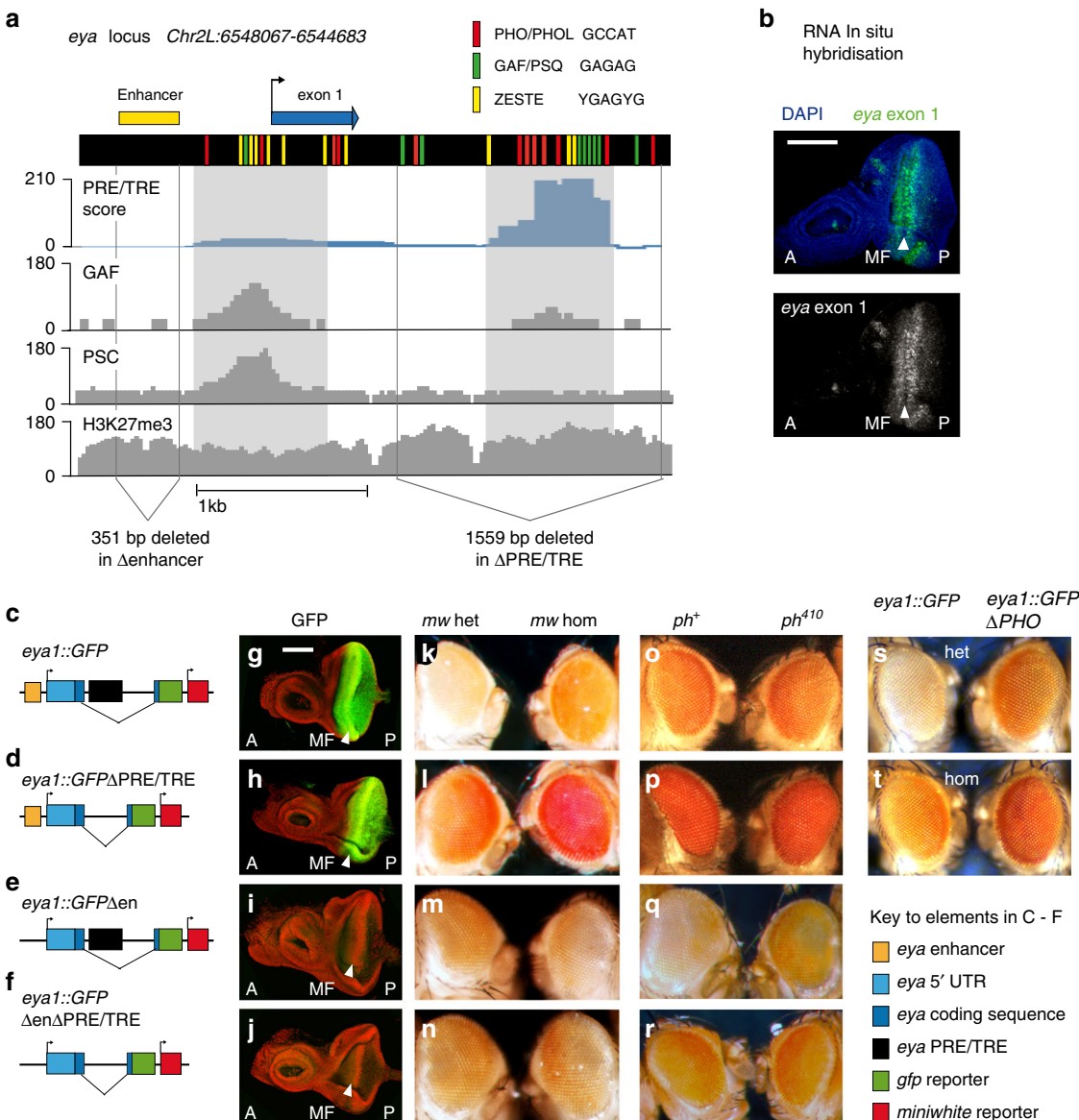

**Fig. 5 Reporter constructs for *eya* PRE/TRE and enhancer. a** 3.4 kb of *eyes absent* (*eya)_* locus included in the *eya* 1 transgene. BDGP Drosophila melanogaster genome version R5/dm3, Chr2L:6548067–6544683. Coloured bars show motif occurrence as indicated. PRE/TRE score (blue track) was calculated according to ref. [12]. Grey: ChIP-seq tracks for the proteins or histone modifications indicated, obtained from ModEncode (http://compbio.med. harvard.edu/modencode/webpage/Chromatin.v0.6.html#ChIP-seq%20and%20ChIP-chip%20data). ChIP-seq 14–16 h Oregon R embryos, Karpen lab. H3K27me3 (track ID 3955), PSC, Posterior Sex Combs (track ID 3960), GAF, GAGA Factor (track ID 4119). Below the plots the regions that are deleted in transgenic reporters are indicated: 1559 bp deleted in ΔPRE/TRE: Chr 2L:6546243–6544711. 351 bp deleted in Δenhancer lines Chr 2L: 6547869–6547519. **b** RNA in situ hybridisation against exon 1 of *eya* on wild type 3rd instar eye-antenna imaginal disc. Anterior (A) posterior (P). Morphogenetic furrow (MF). Scale bar, 100 μm, same for (**g–j**). **c** *eya1::GFP* reporter construct. Sequence encoding *turboGFP* (green) was fused to exon 2 of the *eya* gene (dark blue) in a transgenic reporter construct carrying the *eya* enhancer (yellow), exon 1 (blue) and the intronic *eya* PRE/TRE (black). The sequence shown in (**a**) corresponds to the enhancer, first exon and intronic PRE/TRE. The construct contains a *mw* reporter (red) for detection of expression in adult eyes. **d–f** Variants of the reporter construct lacking the intronic PRE/TRE (**d**), lacking the enhancer (**e**), or lacking both (**f**), were integrated by ΦC31 integration. **g–j** Live imaging of GFP protein in eye antennal discs of 3rd instar larvae homozygous for the transgenes as indicated. Green: GFP; red: SYTO Red live DNA stain. Six to 10 discs per genotype were analysed, giving similar results. **k–n** Eye colours of 4 day-old adult female flies heterozygous (left) or homozygous (right) for the transgenes as indicated. **o–r** Females homozygous for the transgene variants in a heterozygous mutant background for the Polycomb group gene *polyhomeotic* (*ph*). **s, t** *eya1::GFP ΔPHO*: all eight PHO binding sites of the *eya1::GFP* transgene were mutated (see Supplementary Fig. 7D). Four-day old females heterozygote (S) and homozygote (T) for the transgenes are shown.

introduced only at the onset of *eya* activation, and was 3-fold weaker than that required for memory in the previous simulations (Supplementary Fig. 9). Earlier and stronger coupling resulted in silencing of the PRE/TRE in response to the early silent state of the promoter (Supplementary Fig. 9). Second, the best fit was obtained when the values of $p3$ and $p4$ (feedback) were reduced from 0.25 to 0.2, and the feedback-independent transition parameter, $p5$, was increased approximately four-fold to 0.17. The large increase in $p5$ and concomitant reduction in ($p3$, $p4$) renders the PRE/TRE more flexible, allowing it to respond continuously to the promoter state. Thus with specific parameter changes the simple model is able to quantitatively

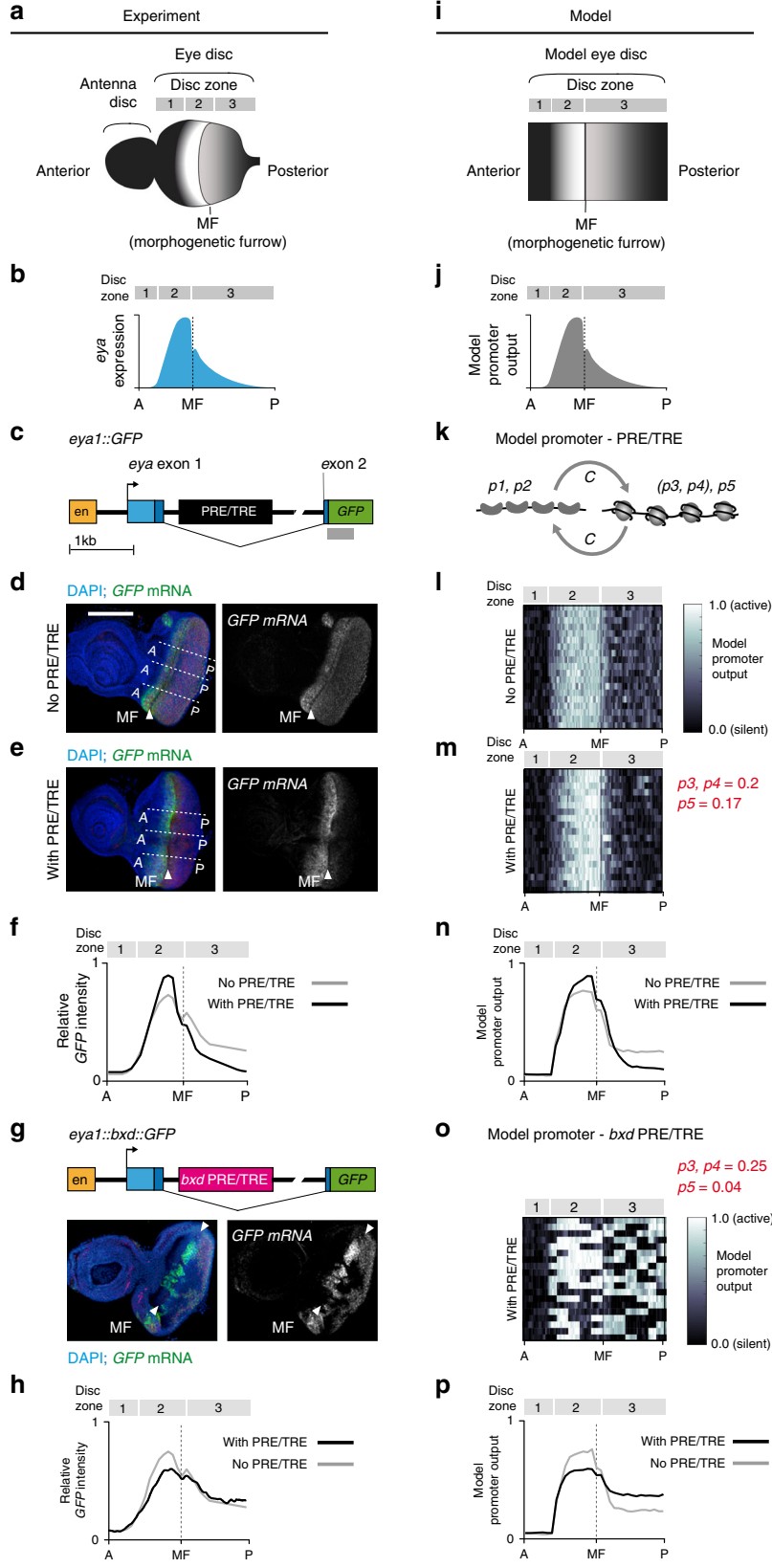

recapitulate the observed behaviour of the *eya* PRE/TRE in the developing eye disc.

Remarkably, when searching for parameters that best reflected the behaviour of the *bxd* PRE/TRE in this assay we found that an optimal fit was obtained with the same coupling parameters as for

*eya*, and for all other parameters (*p3*, *p4* and *p5*), identical values to those determined for the memory assay (Fig. 6o, p and Supplementary Fig. 9). We note that in the posterior part of the disc, the model predicted a higher average expression level than was observed in the experiment (compare Fig. 5h and p). This

**Fig. 6 Two PRE/TREs respond differently to a dynamically regulated promoter. a** Third instar larval eye-antenna disc. **b** *eya* expression levels in mid-3rd instar larval eye disc across zones 1–3 (A, anterior; P, posterior,). **c** *eya1::GFP* reporter construct. Yellow (en), *eya* enhancer; blue, *eya* exons (light blue, 5′ UTR; dark blue, coding region); black, *eya* PRE/TRE. Grey bar: RNA in situ probe used in (**d**) and (**e**). **d, e** RNA in situ hybridisation against *GFP* on mid 3$^{rd}$ instar eye-antenna disc of larvae homozygous for the *eya1::GFP* transgene (**e**), or the same transgene lacking the PRE/TRE (**d**). Scale bar, 100 μm, same for (**g**). Six to ten discs for each transgene gave similar results. **f** Signal intensity profiles along the A-P axis (dotted lines shown on the discs in **d, e**). Multiple line scans perpendicular to the morphogenetic furrow were averaged for six - 10 discs for each transgene and aligned to the morphogenetic furrow, the anterior limit of the expression domain and the posterior edge of the disc. Vertical scale represents signal intensity relative to the maximum of *GFP* mRNA. **g** Top: *eya1::bxd::GFP* reporter construct. 1.5 kb of the *eya* PRE/TRE was replaced in the reporter construct with 1.5 kb of the *bxd* PRE/TRE (see "Methods"). Bottom: RNA in situ hybridization against *GFP* mRNA shows variegation. Similar results were obtained for eight discs. **h** Average signal intensity profiles generated as in (**f**), except that the whole disc area was scanned to reduce noise due to variegation. **i** Model eye disc. The eye disc was modeled as described in Supplementary Methods and Supplementary Fig. 8. **j** The model promoter output was quantified by averaging the anterior-posterior profiles of 100 simulations. **k–m** Model promoter and PRE/TRE. **l** No PRE/TRE. **m** With PRE/TRE (Model 1, $C_{i,m} = 0$ (zone 1); $C_{eya} = 2.5$ (zones 2 and 3)). **n** Average model promoter output for 100 independent simulations. **o** Simulated eye disc showing best fit to *eya1::bxd::GFP* data shown in (**h**). **p** Average model promoter output for 100 independent simulations. See also Supplementary Figs. 1, 7 and 8.

may be due to the large variation in degree of variegation in the imaged eye discs. Nevertheless, the criteria for a good fit, namely the shape of the gradient and the variegating pattern, were fulfilled by a range of values for the parameters *p3, p4* and *p5* that are mutually exclusive to those that fit the smooth *eya* PRE/TRE gradient (Supplementary Fig. 9).

The parameter *p5* (feedback-independent transitions) for the best fit of the *bxd* PRE/TRE model to the data (*p5* = 0.04) was four-fold lower than for the *eya* PRE/TRE model (*p5* = 0.17). The best- fit values of the feedback parameters (*p3, p4*) were higher for the *bxd* PRE/TRE ((*p3, p4*) = 0.25) than for the *eya* PRE/TRE ((*p3, p4*) = 0.2). Low values of *p5* and high values of (*p3, p4*) tend to make the system more bistable and inflexible, leading to variegation, caused by the perpetuation of random active and silent PRE/TRE states that were adopted before coupling. The gradient imposed by the promoter was nevertheless discernable in the *bxd* model at intermediate coupling strengths (Supplementary Fig. 9D). At higher coupling strengths, the variegation imposed by the PRE/TRE dominated completely and the gradient was lost (Supplementary Fig. 9F, G). Taken together these results demonstrate that defined parameter changes enable the simple model to capture both the late switching, fine-tuning behaviour of the *eya* PRE/TRE and the variegation of the *bxd* PRE.

## Discussion

We have combined mathematical modelling and quantitative experimental analysis to examine different modes of PcG/TrxG-mediated gene regulation in *Drosophila* and how they are modulated during development. The unique feature of the model described here in comparison to previous models of chromatin-mediated epigenetic regulation[21,22,31,42] is that chromatin states and transcriptional states are considered explicitly and independently from each other, with regulated coupling between them. These features were essential for the model to recapitulate the observed data, thus identifying the simplest model required to describe the phenomena of interest[43]. We identify specific roles for cell cycle length, transcriptional input, PRE/TRE-promoter coupling and PRE/TRE identity in determining system behaviour. This simple model system is able to generate a rich repertoire of outputs including memory, variegation and fine-tuning and thus provides a unifying theoretical framework for profoundly different modes of PcG/TrxG-mediated gene regulation (Fig. 7).

An important finding of this study is that the model system becomes increasingly stable as development proceeds (Fig. 7a). During the first 13 rapid division cycles the frequent disruptions of replication result in a naive PRE/TRE state in the model. In the initiation phase (cycle 14), in the absence of replication, this naive PRE/TRE stabilises into active and silent states, and can still receive information from a coupled promoter. Finally, in the

maintenance phase, PRE/TRE states become fixed and are imposed on a coupled promoter, giving epigenetic memory (Figs. 2 and 3). In the model, these three phases depend only on cell cycle length and timing of coupling, and no other developmental change in PRE/TRE properties is required to recapitulate experimental data for epigenetic memory.

The dependence of stability of epigenetic memory on cell cycle length has been observed in several theoretical studies[21,31]. Here, we extend those observations to model the entire lifespan of a developing animal. We propose that developmentally regulated changes in cell cycle length may contribute to the observed developmental increase in stability of chromatin states in *Drosophila*[3,6,44–47] and to the progressive accumulation of histone modifications during development[34,35]. In ref. [31] this idea was proposed but not explicitly tested. Interestingly, a similar effect has been observed for heterochromatin in *Drosophila*. HP1 and H3K9 methylation progressively accumulate with the progressive lengthening of S-phase during cycles 11–13[32,33]. This accumulation is in increased upon arrest of cycle 13[33], and abrogated in *grp* mutants, which have shortened cycles 11–13[48].

The fact that the system becomes globally more stable as development proceeds raises the question of how individual genes locally escape this constraint to allow late switching and dynamic responses to promoter input (Fig. 7b). We show here theoretically that the two most important factors enabling this escape are the coupling between PRE/TRE and promoter, and the inherent properties of the PRE/TRE itself.

In the model, regulated coupling between the PRE/TRE and promoter, and the absence of coupling during cycles 1–13, was essential to recapitulate all three data sets (memory of silencing, memory of activation, and the *eya* gradient). This suggests that in the living animal, coupling and decoupling between PRE/TREs and promoters or enhancers may be developmentally regulated, locus-specific and biologically important. Several candidate mechanisms for coupling exist, including spreading of marks from a nucleation site[49], specific PRE/TRE-promoter communication[19,50], regulation by non-coding RNAs[20,51], long-range chromatin interactions[52] and *trans*-regulation by homologous pairing[53,54] (see Supplementary Discussion). Interestingly, several of these phenomena are absent during the early cycles and accumulate during cycle 14, the time window in which we predict that coupling is required[19,50,52].

We have shown both experimentally and theoretically that the identity of the PRE/TRE itself is decisive in determining the regulatory properties of the system. We show experimentally that when two different PRE/TREs are placed in an otherwise identical regulatory setting, the *bxd* PRE/TRE causes variegation whilst the *eya* PRE/TRE gives a smooth gradient. Since the reporters are placed in identical genomic locations, these differences in

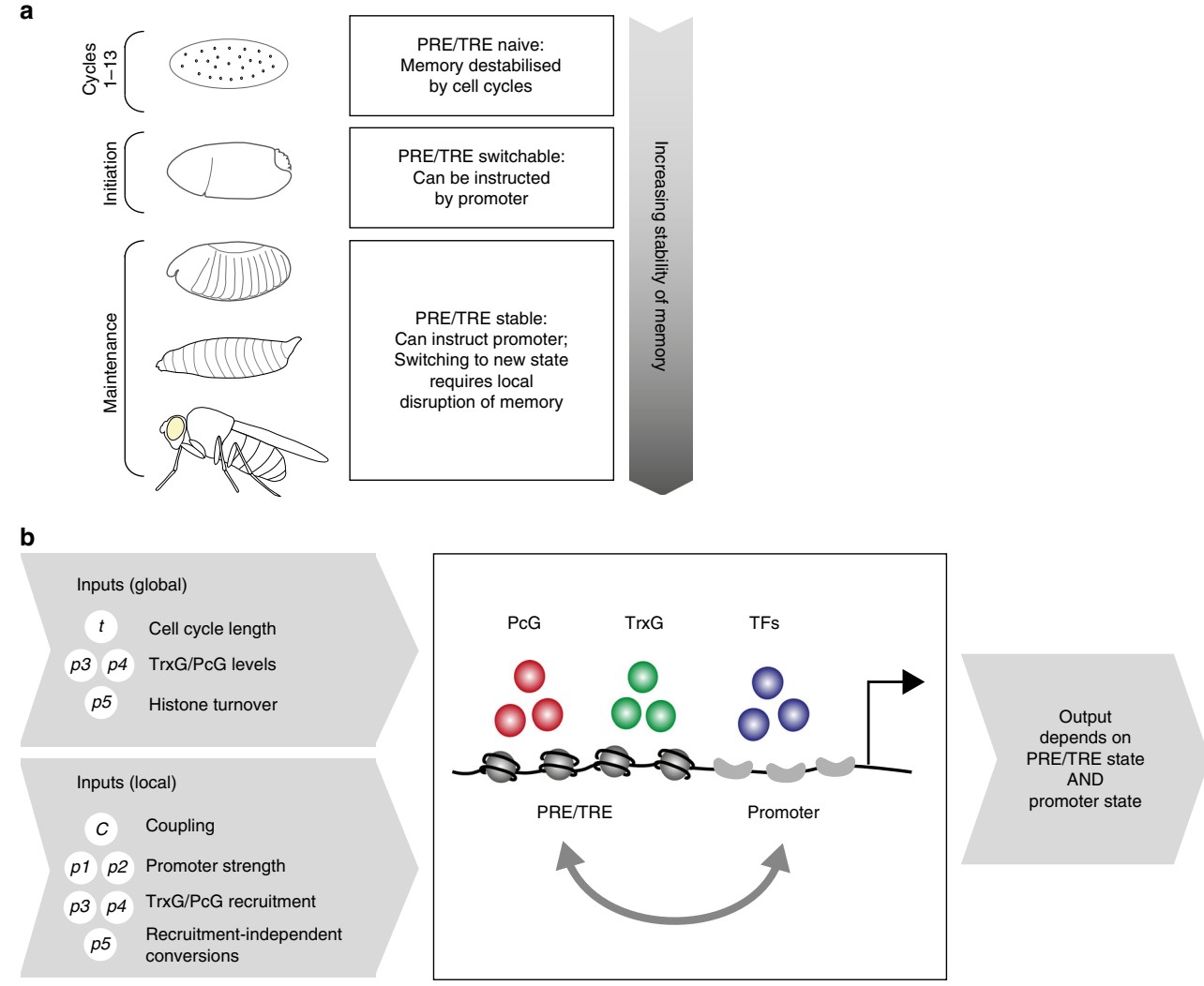

**Fig. 7 Summary of model and main conclusions. a** Summary of developmental changes in stability of memory in the model. **b** Summary of the model and its parameters. PcG: Polycomb group proteins; TrxG: Trithorax group proteins; TFs: transcription factors. Global inputs are those that are expected to affect all PRE/TRE regulated genes in the genome. Local inputs are those that may be regulated specifically for different loci. The output of the model in terms of transcriptional activity depends on the interplay between PRE/TRE state and promoter state. See main text for details.

regulatory output must depend on nucleic acid sequence differences between the two PRE/TREs. In our simple model, the difference between the two PRE/TREs depends on differences in only two parameters: ($p3$, $p4$) (feedback) and $p5$ (feedback-independent transitions).

In the model, the *eya* PRE/TRE with high $p5$ and low ($p3$, $p4$) is flexible and able to respond continuously to promoter inputs, but unable to maintain memory. In contrast, the *bxd* PRE/TRE with a low $p5$ and high ($p3$, $p4$) is more bistable, less flexible and more able to maintain memory of initial states. In the model, if such a strongly bistable PRE/TRE is coupled to a promoter during the initiation phase, it responds to the promoter and later gives stable memory. However, if this same strongly bistable PRE/TRE is coupled to a promoter at later stages, it causes the promoter to variegate. Thus the model predicts that variegation (frequently observed in PRE/TRE reporter assays[3,55], but rarely for endogenous genes in the developing fly) is a result of the late coupling of a strongly bistable PRE/TRE to a promoter. These observations provide a coherent hypothesis for the mechanistic and developmental relationship between variegation and epigenetic memory.

What might be the molecular basis of the difference between the *eya* and *bxd* PRE/TREs? Interestingly, one of the largest differences in known motifs between the *bxd* and *eya* PRE/TREs is the configuration of binding sites for the TrxG protein GAGA factor (GAF) (Supplementary Fig. 7A–C). Whereas the *bxd* PRE/TRE contains more single instances of the GAF binding motif (GAGAG), the *eya* PRE/TRE contains a run of five consecutive GAF motifs (Supplementary Fig. 7A). The GAF protein binds its target sites cooperatively and has nucleosome remodelling activity[56,57]. We propose that the configuration of GAF binding sites in the *eya* PRE/TRE may contribute to increased nucleosome remodelling and thus to reduced stability of chromatin states. The identification of model parameters and defined DNA sequences that cause large differences in output will allow the design of precise perturbation experiments in future to address mechanistic questions linking regulatory output to DNA sequence features. PRE/TRE sequences are complex, diverse, and evolve rapidly[12,24,58]. To date, research has focused on the role of PRE/TRE sequence motifs in recruiting PcG and TrxG proteins[24,25]. We propose that PRE/TRE sequence may also influence the stability of PRE/TRE states, bringing a fresh perspective to the PRE/TRE code.

In conclusion, this work provides a general theoretical framework for PcG/TrxG regulation and systematically extends

concepts of regulation beyond epigenetic memory. The model allows a quantitative dissection of the interplay between chromatin states and transcriptional states, the relationship between PRE/TRE sequence and regulatory output, and the capacity of the system to respond to developmental and environmental signals. Thus this work has broad implications for understanding genome-wide PcG/TrxG regulation throughout development, and the consequences of its disruption in disease.

## Methods

**Mathematical modelling**. See Supplementary Methods.

**Dot blot**. Peptides were purchased from Jpt Innovative Peptide Solutions in unbiotinylated form. Dot blots were performed using 2 µl of the different histone peptides (Supplementary Table 3) spotted in different quantities (300 pmol, 150 pmol, 75 pmol and 37 pmol in PBS) onto a nitrocellulose membrane (Thermo Fisher Scientific). The membrane was dried for 1 h at room temperature before blocking with 5% BSA in TBST (20 mM Tris, pH 7.5, 150 mM NaCl, 0.05% Tween20) and incubation with primary antibody in the same buffer, each for 30 min at room temperature with gentle shaking. The membrane was extensively washed in TBST before incubation for 30 min with secondary antibody in TBST, 5% BSA, followed by extensive TBST and finally, TBS washes. See also[59]. Primary and secondary antibodies (Supplementary Tables 4 and 5) were used at 1:1000 dilution. Signals were visualized by enhanced chemiluminescence (ECL, Thermo Fisher Scientific) and quantified using the gel analysis function in ImageJ (NIH).

**Immunofluorescence on *Drosophila* embryos**. Embryos were collected, fixed and stained according to ref. [60], and stored in methanol at −20 °C. Primary and secondary antibodies (Supplementary Tables 4 and 5) were diluted 1:5000, with the exception of αH3, which was diluted 1:1000. After washing, the embryos were incubated overnight in ProLong Gold containing DAPI (Life Technologies). Embryos were imaged using a Leica DMi8 inverted microscope. Images were acquired with a HC PL Fluotar ×20 dry objective with numerical aperture of 0.55. For deconvolution of the images, the standard express, unsupervised deconvolution option in the Huygens Essential Software 16.05 was used. Image analysis was carried out using Fiji (ImageJ) v2.0.0.-rc_41, and Adobe Photoshop 2015.01.

**Chromatin immunoprecipitation (ChIP)**. ChIP and chromatin preparation on whole embryos was performed as in refs. [58,61] and as follows: Before embryo collection, a pre-laying step of several hours was included, to ensure that only fresh embryos were collected. Embryos of different ages (0–2 h, 2–4 h, 4–6 h, 6–8 h and 8–10 h) were collected, by allowing flies to lay for 2 h and ageing collections for the appropriate time period at 25 °C. For each collection, 1 g of embryos was dechorionated in 3% bleach (2.8% hypochloride) in eggwash (0.7% NaCl, 0.03% Triton X-100) and washed extensively with eggwash, then once with PBS, 0.01% Triton. Embryos were transferred to crosslinking solution (9.5 ml of [50 mM Hepes pH8, 1 mM EDTA, 0.5 mM EGTA, 100 mM NaCl], 30 ml of n-heptane, 0.5 ml of 37% formaldehyde) and incubated with vigorous horizontal shaking for 15 minutes at room temperature. Embryos were transferred to stop solution (0.01% Triton X-100, 125 mM glycine in PBS), then subsequently washed for 10 min with rotation in wash solution A (10 mM Hepes, pH7.6, 10 mM EDTA, 0.5 mM EGTA, 0.25% Triton X-100), and wash solution B (10 mM Hepes, pH7.6, 200 mM NaCl, 1 mM EDTA, 0.5 mM EGTA, 0.01% Triton X-100). Crosslinked embryos were resuspended in 5.5 ml sonication buffer (4 mM EDTA, 50 mM Hepes, pH 7.6, 300 mM NaCl, with addition of Protease inhibitors (Roche Complete, 1 tablet per 30 ml buffer), and 1 ml aliquots were sheared using a Covaris S220 ultra sonicator with the following settings: 65 Watt, 20% DF, 200 cycles per burst. The time of sonication was optimised to obtain chromatin fragments of ca. 500 bp as follows: 0–2 h or 2–4 h old embryos: 45 min; 4–6 h old embryos: 30 min; 6–8 h old embryos: 25 min; 8–10 h old embryos: 20 min. The DNA content of a 50 µL aliquot of each chromatin preparation was measured by spectrophotometry (Denovix) after reverse crosslink and DNA extraction. The chromatin preparations were diluted to equivalent DNA concentrations. Two independent chromatin preparations and two to three independent ChIP assays were performed for each time point. IPs were performed using 50 µg DNA in 300 µl volume with 4.5 µg of each primary antibody as listed in Supplementary Table 4. IPs were performed in 300 µl volume overnight at 4 °C in RIPA buffer (10 mM Tris, pH8.0, 140 mM NaCl, 1 mM EDTA, 1% Triton X-100, 0.1% SDS, 0.1% Na deoxycholate, 1 mM PMSF). Immunocomplexes were purified using Protein A Sepharose beads (CL4B, Amersham, cat # 17–0780–01) (60 µl per IP).

**Linker ligation and qPCR to quantify ChIP enrichments**. Global ChIP enrichments were quantified by linker ligation followed by qPCR as follows (see also refs. [58,61]): 7 µL IP or input material was rendered blunt-ended in 25 µL total volume using the Quick Blunting Kit (NEB). Double-stranded linkers were prepared by annealing 20 mer and 24 mer oligos (see Supplementary Table 6). Prior to annealing, the 24 mer oligo was 5′ phosphorylated using T4 Polynucleotide Kinase

(Roche). 7 µL of the blunted IP material was ligated to a molar excess of double-stranded linkers (1 µM final concentration linkers in 10 µL reaction volume). The linkers are designed so that the annealed 20 mer and 24 mer can only ligate via their blunt end to a blunt-ended fragment. The blunt end of the annealed linkers carries a 5′ phosphate (because the 24 mer was phosphorylated). The overhanging 4 bp do not allow self-annealing. Thus each blunt-ended ChIP fragment can only be ligated to one linker on each end.

Linker-mediated ChIP amplification is normally used with 10–15 PCR cycles to amplify material prior to site-specific qPCR. In contrast, for whole-genome qPCR, samples were not amplified after linker ligation but were subjected directly to real-time PCR using Sso Advanced SYBR Green Supermix (BioRad) and the 20 mer oligo as primer at 150 nM final concentration, with the following program parameters: 2 min 50 °C, 2 min 94 °C, 40 × [15 s 94 °C,1 min 60 °C, 1 min 72 °C].

As a control for linker ligation, a gel-purified 448 bp blunt-ended PvuII fragment of pBluescriptIIKS+ was ligated to linkers under the same conditions as the ChIP ligations, and a dilution series of known quantities was subjected to qPCR, showing that linker ligation is efficient and amplification is linear with fragment concentration. Primer efficiency was calculated on the basis of plasmid and input dilution series to be 2.0 ± 0.3 (meaning that the product is amplified 2.0 ± 0.3 times per cycle). To further control for uniform amplification by qPCR we compared input chromatin samples before and after blunting, ligation and qPCR by agarose gel electrophoresis. This showed a similar size distribution of fragments around the mean of 500 bp, thus the linker-mediated qPCR does not appear to introduce bias for a particular fragment size.

ChIP and input samples from two independent chromatin preparations were subjected to qPCR in the same plate, each in a 5-fold dilution series (1 µl, 0.2 µl and 0.04 µl of the ligation reaction or of a 1:100 dilution of the input chromatin). We compared technical replicates by evaluating the correlation between data series from ChIP and inputs from independent chromatin preparations from each time point. We observed a high correlation between independent ChIP assays performed with the same antibody from the same time point (Supplementary Fig. 6).

We calculated enrichments for each modification and for unmodified histone H3 relative to (input/100) for each dilution using the following formula: enrichment relative to input = $2^{(Ct\ input\ –\ Ct\ ChIP)}$. (Ct = the cycle number at which the signal crosses the threshold). This gives enrichments as a proportion of input, where an enrichment of 1 would be obtained if the quantity of IP material were equivalent to 1% of the input (because the input samples were diluted 100-fold prior to qPCR). This dilution step was necessary to give input Ct values in a similar range to those of the ChIP samples. The ChIP vs. input calculation gave a measure of the reproducibility of ChIP assays (see also Supplementary Fig. 6). To calculate the enrichment of each modification relative to Histone H3 ChIP one could use (2$^{(Ct\ input\ –\ Ct\ ChIPx)}$)/(2$^{(Ct\ input\ –\ Ct\ H3\ ChIP)}$). However, this expression simplifies to 2$^{(Ct\ ChIP\ H3\ –\ Ct\ ChIPx)}$, and thus the input is not required in this case. Indeed we found that including the input in the former calculation led to amplification of errors. Thus for the data shown in Fig. 4e and Supplementary Fig. 6C, the enrichment of each modified histone or PC was calculated as a proportion of the H3 ChIP using the latter calculation. The H3 ChIP samples were not diluted, and thus a relative enrichment of 1 would be obtained if the quantity of IP material from ChIP x is equivalent to 100% of the IP material from the H3 ChIP. We note that such a result would not imply that 100% of H3 molecules carry the modification in question, because the antibodies used are very unlikely to have identical affinities for their epitopes. In ref. [35] the absolute numbers of modifications are calculated as a percentage of total H3 molecules, which is a different calculation than the one we perform here. To avoid confusion, we use the term "enrichment relative to Histone H3 ChIP" on the axis of Fig. 4e. The vertical scale indicates the result of the calculation 2$^{(Ct\ ChIP\ H3\ –\ Ct\ ChIP\ x)}$.

**Generation of *eya1::GFP* reporter construct and variants**. The *eya1::GFP* reporter construct was generated using primers and templates as shown in Supplementary Table 6. All PCR steps were performed using Phusion polymerase (NEB). Two genomic fragments A and B of the *eya* locus (see Fig. 5) were each cloned separately into pCRII TOPO (Invitrogen) and sequenced. Fragment A (2L:6544449–6548972, flybase version FB2010_05; *D.mel* genome version R5.28) contains the eye specific enhancer, the first exon and the *eya* PRE/TRE (see Fig. 6). Fragment B (2L:6530964–6535677) consists of intronic sequence including a branch site and the first three codons of exon 2 in order to enable splicing. Fragment B was fused to a fragment encoding TurboGFP (fragment C) using overlap extension PCR[62] (Supplementary Table 6). The fused product B,C was cloned into pCRII TOPO and sequenced. Fragment A was cloned with EcoRI into pCRII TOPO BC, and the fused product was cloned into the EcoRI/ XhoI cut pKC27_mw vector[41], resulting in the final *eya1::GFP* construct. The *eya1::GFP* deletion constructs were generated by overlap extension PCR[62] using the *eya1::GFP* construct as a template, and primers as shown in Supplementary Table 6. Details of cloning are given in the legend to Supplementary Table 6. For the *bxd* replacement construct, *eya* PRE/TRE coordinates Chr2L:6546243–6544711, were replaced with *bxd* PRE/TRE coordinates Chr3R:12590916–12589364, *D.mel* genome BDGP R5/ dm3. *eya1::GFP* was modified to replace the *eya* PRE/TRE with four unique cloning sites by replacing a 1982bp FspAI/Bsu36I fragment containing the PRE/TRE with a synthetic 569 bp FspAI/Bsu36I fragment (Mr.Gene http://mrgene.com), carrying NheI, AvrII, KpnI and SacI sites in place of the PRE/TRE sequence. The *bxd* PRE/

TRE (see Fig. 6, Supplementary Fig. 7) was amplified from genomic DNA using primers shown in Supplementary Table 6, and was cloned with NheI/KpnI into this modified *eya1::GFP* vector. The eya PRE/TRE with mutated PHO sites was synthesised as a 1982bp fragment (Mr.Gene http://mrgene.com) with mutations as shown in Supplementary Fig. 7D and legend. All constructs were sequenced prior to injection.

**Generation and genetic analysis of transgenic flies**. Transgenic fly lines carrying *eya1::GFP* or variants were generated by ΦC31 mediated integration by the IMBA fly injection service. The landing site used here corresponds to site 2 at position 46E1 in ref. [41] (chr. 2R) at genomic location 5,965,083 (flybase version FB2010_05; *D.mel* genome version R5.28). All fly stocks are available on request. To test the effects of the $ph^{410}$ mutation on *miniwhite* expression, flies of the following genotypes were crossed for each transgene: females: $w^-$, $ph^{410}/ w^-$, $ph^{410}$;Transgene/ CyO x males: $w^-$; Transgene/CyO. Female progeny lacking the CyO balancer were heterozygous for the $ph^{410}$ mutation and homozygous for the transgene. These flies were aged for four days before taking photographs. The $ph^{410}$ allele (FBal0013768) is a hypomorphic allele of *ph*. It disrupts the proximal *ph* transcription unit but leaves the closely related distal transcription unit intact.

**Live GFP imaging**. 3rd instar larval eye discs were stained in 5 μM SYTO Red Fluorescent Nucleic Acid Stain (Invitrogen) and imaged on a confocal microscope LSM 700/Axioimager (Zeiss).

**In situ hybridisation**. PCR products were generated for two in situ probes (*eya* exon 1 and GFP) using the primers and templates shown in Supplementary Table 6. PCR products were cloned into pCRII vector (Invitrogen), sequenced, and checked for the correct orientation. For in situ hybridization, probe sequences were PCR amplified from pCRII using M13 primers such that the PCR products contained the T7 promoter. Probes were then in vitro transcribed using T7 polymerase (Roche). The probe was labelled with fluorescein (Roche), detected with primary antibody: Mouse anti-fluorescein (Roche), secondary antibody: goat anti-mouse-HRP (Invitrogen) and visualised with Alexa Fluor 488 Tyramide (Invitrogen) using Tyramide Signal Amplification kit (TSA™, Invitrogen) according to manufacturers instructions. Tissues were mounted in Vectashield with DAPI (Vector Laboratories) on microscope slides. Images were taken using confocal microscopy, with LSM 700 Axioimager (Zeiss).

Signal intensity profiles were generated by evaluation of maximum intensity projections using the line scan function of the imaging software MetaMorph (Version 7.1.1.0). The line scans were placed in anterior-posterior orientation at 90° to the morphogenetic furrow as shown in Fig. 6d, e. Three to four scans were performed for each disc. Average Y-values of each channel (Red/Green/Blue) were obtained by applying a scan width of 50 pixels. Line scans from six to ten individual eye discs of similar size from each experiment were aligned with respect to the position of the morphogenetic furrow and the posterior disc edge on the X-axis, and averaged for the plots shown in Fig. 6f. For vertical scaling the average line scans were normalised by setting the maximum measured intensity of each disc to 1.

**Reporting summary**. Further information on research design is available in the Nature Research Reporting Summary linked to this article.

## Data availability

The ChIP-seq data shown in Fig. 5a are publically available at [http://compbio.med. harvard.edu/modencode/webpage/Chromatin.v0.6.html#ChIP-seq%20and%20ChIP-chip %20data]. H3K27me3 (track ID 3955), PSC, Posterior Sex Combs (track ID 3960), GAF, GAGA Factor (track ID 4119). All other relevant data supporting the key findings of this study are available within the article and its Supplementary Information files or from the corresponding author upon reasonable request. Source data for Fig. 4e are provided with this paper. A reporting summary for this Article is available as a Supplementary Information file. Source data are provided with this paper.

## Code availability

The computer code used in this study is available as supplementary software and has been deposited at [https://github.com/Ringrose546/Beyond_memory_V2].

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

## Acknowledgements

This work was supported by Austrian Academy of Sciences http://www.oeaw.ac.at/ (L.R. and F.R.) the EU FP7 Network of Excellence Epigenesys (FP7-HEALTH-F4-2010-257082) http://www.epigenesys.eu/ (L.R. and F.R.), the DFG Exzellenzinitiativ IRI-Lifesciences https://www.iri-ls.hu-berlin.de/en (L.R. and J.R.), the EU H2020-MSCA-ITN PEP-NET (Project number 813282) and by the BBSRC Institute Strategic Programmes GRO (BB/J004588/1) and GEN (BB/P013511/1) (M.H.). We acknowledge support by the Open Access Publication Fund of Humboldt-Universität zu Berlin.

## Author contributions

L.R. conceived the study, performed modelling and wrote the computer code and the paper. M.H. hosted L.R. during the initial stages of the project, provided advice on modelling and contributed to the paper. J.R. performed experiments shown in Fig. 4 and Supplementary Figs. 4, 5 and 6. F.R. performed experiments shown in Figs. 5, 6 and Supplementary Fig. 7. L.R. supervised J.R. and F.R.

## Funding

## Competing interests

The authors declare no competing interests.
