## [Peer Review File · Nature Communications]

Reviewers' comments:

Reviewer #1 (Remarks to the Author):

A theoretical model of Polycomb/Trithorax action unites stable epigenetic memory and dynamic regulation

Jeannette Reinig, Frank Ruge, Martin Howard and Leonie Ringrose

This is a very significant paper, that for the first time brings together a rigorous mathematical model of epigenetic memory and experiments in the best characterized metazoan system for testing the function of cell memory elements. In particular, the modelling and experiments show very nicely how localized memory elements can be utilized to provide either stable cell memory or a sharpening of the response to transcriptional regulation. In addition, the modelling and experiments show the potential for rapid replication cycles to destabilize chromatin-based memory. These are important findings of broad applicability and interest.

The basic model has the attraction of being reasonably simple and yet powerful. A very simple model for promoter regulation by TFs is combined with a well-established nucleosome modification positive feedback model for the PRE/TRE. The novel feature is the symmetrical coupling of the TF state of the promoter and the modification state of the PRE/TRE. The mechanism of this coupling is also simple and feasible, with the state of the PRE/TRE affecting TF affinity, and TF occupation affecting the rates of recruited nucleosome conversion. Coupling introduces a new level of positive feedback where, for example, a low activity PRE/TRE fosters lower promoter activity, which in turn fosters lower PRE/TRE activity.

The absolute strength of this coupling required for the model to work is not apparent in the main text, and is even difficult to appreciate in the Supplementary, where a $C=2$ case is plotted. The value of $C=8$ for the coupling factor (used in Figs 2 and 3) means that the activity-promoting reactions at the promoter and PRE/TRE can change, by my understanding, over a ~ 3000 -fold range, depending on the state of the system, with the same change inversely applied to the inhibitory reactions. (There is a ~ 12 -fold change for the $C=2.5$ case for Fig 5). Some comment on the required strength of this mechanism and its feasibility seems warranted.

With minimal manipulation of the onset of coupling and of TF availability in time and space, the model is able to reproduce previous observations of PRE/TRE-dependent memory of embryonic TF distribution patterns. The model is also able to explain the PRE/TRE-dependent spatial 'sharpening' of *eya* expression across the eye disc.

Importantly, two predictions of the modelling are tested. (1) A relative lack of modifications during the rapid replication cycles of early development and an increase in modifications as cells move to slower replication cycles

is shown by in situ hybridization and ChIP assays. (2) Replacement of the *eya* PRE/TRE with the memory-competent *bx*d PRE/TRE leads to *eya* expression variegation in the eye disc.

There are some instances where the model and experiment show minor divergences, which is not surprising for a simple model and a complex system, but it might reassure readers if some comment on these is made in the text:

(1) In Fig 3D, there is a difference between the model and the experiment that is not discussed, where a late heatshock gives *lacZ* expression but no increase is seen in the model (due to suppression by the M state of the PRE/TRE). I guess one could imagine a more complex model with some promoter activity that is not coupled to the PRE/TRE.

(2) The timing of the onset of the histone modifications in Fig 4 do not match the model precisely and some comment could be useful. I suppose there could be detection issues, but we also are not

sure exactly how PRE/TRE memory is encoded.

(3) The match of the experiment and model for the bxd PRE/TRE is not so good in the posterior of the eye disc (compare Fig 5H and P).

In general the paper is very well written and presented, and everything is thoroughly explained. However, I have a few suggestions that would have made it easier for me to follow:

In Fig 1C it would aid readers to state "(s=10 sites)" and "(S=40 nucleosomes)" on the figure (or at least in the legend), as these are used throughout, and would save a search in the Supplementary.

In Fig 2 some more explanation and detail of the model could be added by putting a simple table to the right of the 2A panel, showing the change in coupling, p1 and p2 (for A and P). Tick marks on the timeline at the left of 2B could be used to illustrate replication events.

In Fig 4D the bars for PC levels could be white to distinguish them from the chromatin bound modifications. Could the apparent lack of PC in the early cycling nuclei be an alternative explanation for the lack of PRE/TRE memory at this stage?

Ian Dodd

Reviewer #2 (Remarks to the Author):

In their article entitled "A theoretical model of Polycomb/Trithorax action unites stable epigenetic memory and dynamic regulation", the authors propose, by quantitative experiments and theoretical modeling, to decipher the role of PRE/TRE genomic modules in the control of gene expression via the PcG/TrxG system. Consistently with a recent computational investigation of the molecular complexity of PcG/TrxG system (Sneppen & Ringrose Nat. Comm. 2019, in press), they introduce a simple effective 3-states model that somehow extends previous related models (in particular the one introduced by Kim Sneppen's and Martin Howard's groups), but that now explicitly introduces a coupling between the promoter (TF-binding) and the PRE/TRE (chromatin/histone marks) states; a main novelty is that now the epigenetic states and the promoter states are considered separately. When confronting the outputs of their stochastic simulations to experimental data (obtained from several published studies or from experimental assays designed and carried out by the authors for the purpose of this article) they show how a minimal version of the model (3 parameters) can recapitulate various experimental observations associated with different gene regulation logics involving PRE/TRE elements: (I) memory of silencing (II) memory of activation and (III) graded transcriptional response.

A main outcome of this article is the demonstration that the cell cycle duration in the early stages of embryogenesis controls the epigenetic stability of the PRE/TRE states and thus the memory in cis of gene activity: early rapid divisions essentially maintain the PRE/TRE in a naive unmarked state whereas subsequent lengthening of cell cycle drives the system toward bistability (flexibility), where the specific PRE/TRE state is instructed by the corresponding promoter activity and then stably maintained, locking the system in the desired state (fidelity). Such cell cycle duration global constraint was already proposed in previously published pure theoretical studies (Dodd 2007, Zerihun 2016), where it was indeed suggested that stability of epigenetic state during these rapid cell divisions could not be achieved for reasonable parameter values. This present study provides the first clear demonstration of that effect.

Overall, this is a very impressive and elegant study, that manages to combine theoretical modeling and quantitative experiments to provide new and important in-depth understanding of the PcG/TrxG system. The article is well written, very clear and documented, easily accessible to a broad audience ie to scientists from the computational and biological community. Their theoretical and experimental works supporting their demonstrations are very solid and fully convinced me. The results reported in that manuscript open new exciting perspectives in the field of quantitative description of PcG/TrxG-mediated gene regulation and more generally in the field of quantitative epigenetics. This study is a perfect example of how a relatively "simple" theoretical model can provide strong in-depth knowledge when combined with quantitative experiments and thus will undoubtedly encourage further similar collaborative works on related topics.

For all these reasons I recommend the publication of this article in Nature Communication.

I have only few minor concerns:

(1) It is not clear to me how they perform the mapping of the time in their simulation: in other words, how do they fix the 12 s per iteration ?

(2) The experimental evidence of the influence of cell cycle duration in chromatin state establishment and maintenance has been also obtained by O'Farrell's group in the context of constitutive heterochromatin in *Drosophila* (Shermoen et al., *Curr Biol* (2010), Farrel et al., *Annu Rev Genet* (2014), Yuan et al., *Gene Dev* (2016), Seller et al., *Genes Dev.* (2019) ...). I thus suggest to refer to that work since it relies on a very similar concept.

(3) Typos:

- p7, l18; p14, l16; p22 l 18; p28, l6; : extra parenthesis
- p14, l8: "only Supplementary to ..." ????
- p15, l12: comma after the reference 36
- p19,l21: missing parenthesis
- p22,17: missing comma after ref 37
- p23, l24: extra point after ref 58

Reviewer #3 (Remarks to the Author):

The paper by Reining and co-authors reports on development of numerical model of Polycomb/Trithorax action that re-capitulates the ability of *Drosophila* PRE/TREs to install epigenetic memory of both repressed and active transcriptional states. Numerical modelling of epigenetic processes is a hot research field that should be of interest to many readers of Nature Communication and the findings of Reining et al. provide substantial advances. The authors start by formulating the model which behaviour recapitulates classic *Drosophila* experiment. In this experiment the lacZ expression is driven by enhancers of the Ubx gene. During early stages of embryonic development spatial pattern of lacZ expression is defined by interplay of maternally provided repressor and activator proteins that compete for binding to Ubx enhancers. At later stages of development the lacZ reporter becomes active throughout the embryo and larva unless coupled to a PRE/TRE. The authors explore the range of parameters where expression pattern of the reporter simulated by the model match the experiment. They demonstrate that such

values exist thereby showing that a simple model can recapitulate some basic features of epigenetic repression. They also show that rapid cell divisions in the early fly embryo and, perhaps as a direct consequence of this, the lack of mutual influence of PRE/TRE state and transcriptional status of the target gene is, in principle, sufficient to explain the "switch-like" behaviour of Polycomb/Trithorax system in the context of Drosophila homeotic genes.

Reining and co-authors then go on to investigate whether the same numerical model run with the same set of parameters can recapitulate epigenetic memory of activation. Here the authors attempt to simulate experiments by Cavalli and Paro. In these experiments the lacZ and white reporter genes are activated by short pulse of GAL4. Given early during embryonic development, GAL4 activates the reporters, which then remain active throughout development. In contrast, if GAL4 is given at later developmental stages, the reporters get transiently activated but then get re-repressed again. The authors claim their model recapitulates results of Cavalli and Paro. This, at first glance, seems like an overstatement. Yes, the author's model results in the stable expression of the reporters only when the activator is given during early embryonic development. However, in the model, when the activator is given at later stages the reporter is simply not switched on (Figure 3C), which automatically explains the lack of persistent activity. This is different from the results reported by Cavalli and Paro, who made a point of transient activation at later stages without triggering epigenetic memory of active state. Is the model of Reining et al. wrong or the results of Cavalli and Paro a special case? To bolster the support for the model, the authors may want to look at experiments by Poux et al. PMID: 11973279 whose results are, actually, matching those returned by the author's model.

The underlying assumption of the model is that the PRE/TRE-mediated memory of transcriptional state is based on modified histones. This is well demonstrated for Polycomb repression, less so for Trithorax system. The model requires that in the early embryos histone modifications at PRE/TRE remain low. From this the authors "predict" that H3K27me3 (the mark of Polycomb repression) and H3K36me2, H3K36me3, H3K4me1, H3K4me3 (presumed marks of Trithorax system) are low in the early embryos and accumulate slow during development. The authors test their "prediction" by immunostaining of embryos and something they call "ChIP followed by whole-genome qPCR". In my opinion, this set of experiments is out of place. First, the increase in the bulk levels of H3K27 and H3K36 methylation in the developing fly embryos was already observed by mass-spectrometry (Bonaldi et al., 2004 PMID: 15188406) and, for H3K27 and H3K4, by IF and ChIP-seq (Zenk et al., PMID: 28706074). Second, the bulk levels of H3K27me3 are done largely by hit-and-run methylation by non-tethered PRC2 (Lee et al., PMID: 25986499; Ferrari et al., PMID: 24289921). This process is not mechanistically related to H3K27 methylation around PREs and, most likely, has different kinetics. Finally, I do not understand "ChIP followed by whole-genome qPCR" experiments presented on Figure 4E. Do the authors just take all DNA ChIPed with an antibody of interest, amplify it using random primers or adapters and then measure the concentration of amplification products? In this case, why not directly measure the concentration of ChIPed DNA using Q-bit or similar sensitive method and avoid potential amplification biases. In fact, I am puzzled by values shown on Figure 4E. Some of them differ by almost an order of magnitude from those reported by Feller et al., PMID: 25578876 for cultured drosophila cells using well controlled mass-spectrometry.

The paper concludes with an attempt to computationally and experimentally model the developmental regulation of the *eya* gene. The authors claim that at this gene the Polycomb/Trithorax system does not simply maintain the repressed or active transcriptional state but sharpens the gradient of *eya* expression in the eye-antenna imaginal disc. This part I have the greatest difficulty with. The authors say that *eya* gene has an enhancer upstream of the transcription start site (TSS) and a PRE/TRE in the first intron. Therefore they make a reporter construct where they fuse the upstream and 5' parts of the *eya* (fragments A and B) to the GFP open reading frame (Figures 5C, S6). Then they look at GFP expression from this construct as well as its truncated variants lacking the enhancer or the part of the first intron that contains presumptive PRE/TRE. They further compare the GFP expression from these constructs to the GFP

expression from the matching set of constructs where the "PRE/TRE" part of the first intron is replaced with bxd-PRE of the Ubx gene. While the construct with bxd-PRE showed stochastic loss of *eya* expression in the eye-antenna imaginal disc (Figure 5G), the original construct gave GFP expression gradient resembling that of endogenous *eya* expression (Figure 5C). The authors suggest that the two PRE/TREs are inherently different and try to model this. Here is where I get lost. I do not understand why the authors say that *eya* PRE/TRE is in the first intron. From genome-wide mapping (attached is the IGB screen shot with ChIP-seq data from Kahn et al., PMID: 27557709, but modENCODE data shows the same) it appears that there is a major PRC1 and PRC2 binding site just downstream of the *eya* enhancer and just upstream of its TSS. From all we know, that is the major PRE/TRE, which is included in all author's constructs. Even if the first intron contains another weaker PRE/TRE, replacing that region with strong bxd-PRE results in constructs with two major PREs instead of the one. This provides the simplest explanation of why the construct with bxd-PRE shows stochastic loss of eye expression while the original construct does not. Note, although the "PRE/TRE" region of the first intron seems necessary to make sharp *eya* gradient, there is no evidence that its action depends on Polycomb or Trithorax proteins. Unless I overlooked something, I am afraid the authors have misunderstood the organization of *eya* gene and designed their experiments in a way that their results are hard to interpret.

To summarize, I suggest that the authors do some editing of the "The same model recapitulates epigenetic memory of activation", remove the "Chromatin modifications increase from low to high levels during *Drosophila* development as predicted by the model" and "Two PRE/TREs respond differently to a dynamically regulated promoter" parts and revise the discussion accordingly. This being done, I recommend the paper to be published by Nature Communication.

Reviewers' comments:

Reviewer #1 (Remarks to the Author):

A theoretical model of Polycomb/Trithorax action unites stable epigenetic memory and dynamic regulation

Jeannette Reinig, Frank Ruge, Martin Howard and Leonie Ringrose

This is a very significant paper, that for the first time brings together a rigorous mathematical model of epigenetic memory and experiments in the best characterized metazoan system for testing the function of cell memory elements. In particular, the modelling and experiments show very nicely how localized memory elements can be utilized to provide either stable cell memory or a sharpening of the response to transcriptional regulation. In addition, the modelling and experiments show the potential for rapid replication cycles to destabilize chromatin-based memory. These are important findings of broad applicability and interest.

The basic model has the attraction of being reasonably simple and yet powerful. A very simple model for promoter regulation by TFs is combined with a well-established nucleosome modification positive feedback model for the PRE/TRE. The novel feature is the symmetrical coupling of the TF state of the promoter and the modification state of the PRE/TRE. The mechanism of this coupling is also simple and feasible, with the state of the PRE/TRE affecting TF affinity, and TF occupation affecting the rates of recruited nucleosome conversion. Coupling introduces a new level of positive feedback where, for example, a low activity PRE/TRE fosters lower promoter activity, which in turn fosters lower PRE/TRE activity.

Response: We thank this reviewer for thorough reading of the manuscript and for excellent constructive suggestions, which we address below.

Reviewer's specific comments/ requests (our numbering added).

1) The absolute strength of this coupling required for the model to work is not apparent in the main text, and is even difficult to appreciate in the Supplementary, where a $C=2$ case is plotted. The value of $C=8$ for the coupling factor (used in Figs 2 and 3) means that the activity-promoting reactions at the promoter and PRE/TRE can change, by my understanding, over a ~ 3000 -fold range, depending on the state of the system, with the same change inversely applied to the inhibitory reactions. (There is a ~ 12 -fold change for the $C=2.5$ case for Fig 5). Some comment on the required strength of this mechanism and its feasibility seems warranted.

Response: We thank the reviewer for pointing out that this was unclear in the original manuscript, and we have made the following changes to clarify the explanations of coupling:

- a) For equivalent coupling strength in models 1 and 2, the value of C is 2 fold higher in model 1 than in model 2 (as illustrated in Figures S1E and F). In the original version of the paper the examples of coupling given in Figures S1C and D were shown with $C = 2$ for both models, which did not enable a direct comparison and made it difficult to envisage how the values of model parameters were affected in the simulations using the optimum values determined later. We have re-plotted the response curves for both models in Figures S1C and D for $C = 4$ (model 1) and $C = 2$ (model 2), so that the effect of coupling in later simulations can be better understood. We now use these values in Figures 2, 3, S2 and S3, so the plots shown in Figure S1C and D can be directly used to interpret the effect of coupling in those figures (see also response to point 2.1 below).
- b) In Figure S8, we have now included the response curves for the relevant values of C in each case ($C = 2.5$ for model 1 and $C = 1.25$ for model 2). We hope that this will enable the reader to directly relate the value of C to the absolute coupling strength in each case.
- c) We have modified the text on page 7, to include the following explanation in the introduction of the model:

"The exact mechanism of this coupling is not known but may include looping, spreading of chromatin marks or interaction of homologs. These mechanisms are not explicitly included in the model, but the mathematical description of the interaction between PRE/TRE state and promoter, and the strength of this interaction, were varied to identify the model that best fits the data. 14 different coupling models were explored (Figure S1), which differ in the number of model

parameters (p_1 , p_2 and (p_3, p_4)) that are adjusted at each iteration (Figure S1G,H), and the mathematical relationship between PRE/TRE state and promoter state (Figure S1A, B). This analysis showed that only two mathematical descriptions of coupling gave robust results in all tests, designated as models 1 and 2 in the rest of this paper. Models 1 and 2 have in common that they adjust all of the four model parameters at each iteration (p_1 , p_2 and (p_3, p_4)). Model 1 is used for all results shown in the main figures, models 1 and 2 are compared in Figures S2, S3 and S8. Coupling strength in the model is adjusted by the parameter C (Figure 1B), which determines the magnitude of the response of the promoter to a given PRE/TRE state, and that of the response of the PRE/TRE to a given promoter state.”

- d) In addition, on page 10, we have introduced a short description of the results of the fitting for coupling strength:

“A minimum value of $C_{i,m}$ (coupling strength during initiation and maintenance phases) was required for optimum memory (minimum $C_{i,m} = 4$ for model 1, minimum $C_{i,m} = 2$ for model 2, see Figure S2D). These values represent equivalent coupling strengths in both models (Figure S1E, F). At lower $C_{i,m}$ values the system responded too slowly to changes in promoter or PRE/TRE state, and memory was not correctly established during the initiation phase. In general terms, the meaning of C can be understood as the strength of bias towards a given state. Thus for example, for the values of C given above, a fully active promoter will bias a coupled PRE/TRE approximately 55 - fold more towards activation than silencing (see Figure 1E, In $(\text{Factor}_{p4}/\text{Factor}_{p3}) = 4$ for $C = 4$, fully active promoter in model 1). The potential molecular mechanisms by which coupling may be achieved are addressed in the discussion.”

- e) Finally, we refer in the discussion to the required strength of coupling and its feasibility (page 26):

“The strength of coupling required in the model to recapitulate experimental observations depends on the experiment in question. Stronger coupling was required for accurate memory of silencing and activation ($C = 4$, model 1) than to model the dynamic *eya* gradient ($C = 2.5$, model 1). A value of $C = 4$ in model 1 corresponds to a maximum 55 - fold bias of the PRE/TRE or promoter towards one or other extreme state in response to the corresponding extreme state of the other element (Figure S1E). For the *eya* gradient, $C = 2.5$ in model 1 corresponds to a 12 - fold bias (Figure S8).

Although regulated coupling is essential in the model, its molecular meaning in biology is not clear, thus it is difficult to assess whether the coupling strengths given above are feasible in molecular terms. We note however, that a PRE/TRE can give up to 48 – fold repression of a linked reporter gene²², thus large coupling effects do exist, although their molecular mechanisms are unclear.”

Reviewer 1 Continued

With minimal manipulation of the onset of coupling and of TF availability in time and space, the model is able to reproduce previous observations of PRE/TRE-dependent memory of embryonic TF distribution patterns. The model is also able to explain the PRE/TRE-dependent spatial ‘sharpening’ of *eya* expression across the eye disc.

Importantly, two predictions of the modelling are tested. (1) A relative lack of modifications during the rapid replication cycles of early development and an increase in modifications as cells move to slower replication cycles is shown by in situ hybridization and ChIP assays. (2) Replacement of the *eya* PRE/TRE with the memory-competent *bx*d PRE/TRE leads to *eya* expression variegation in the eye disc.

Reviewer’s specific comments/ requests (our numbering added).

2) There are some instances where the model and experiment show minor divergences, which is not surprising for a simple model and a complex system, but it might reassure readers if some comment on these is made in the text:

2. 1) In Fig 3D, there is a difference between the model and the experiment that is not discussed, where a late heatshock gives lacZ expression but no increase is seen in the model (due to suppression by the M state of the PRE/TRE). I guess one could imagine a more complex model with some promoter activity that is not coupled to the PRE/TRE.

Response: We thank the reviewer for drawing our attention to this point, which was also noted by reviewer 3. We agree this is an important issue and have explored the model further, showing that transient activation is in fact possible with reduced coupling strength between the PRE/TRE and the promoter throughout the simulation. It was not necessary to make the model more complex, simply to relax the coupling ($C = 4$ instead of $C = 8$). We have made the following changes to document this:

a) We provide a new version of Figure 3, in which all simulations were performed with $C_{i,m} = 4$ (model 1). In the original Figure 3, $C_{i,m} = 8$. This loosening of coupling results in a transient activation of the reporter during the heatshock (Figure 3D, cycle 15). The other conclusions of the model do not change with this lower coupling strength.

b) Figure S3 now shows models 1 and 2 for $C_{i,m} = 4$.

c) We have modified the text on page 14 to comment on this:

“In particular, a heat shock given after the end of the initiation phase (4h50) allowed transient activation of the reporter but failed to induce stable switching and memory (Figure 3D, late). In contrast, a heat shock given during the initiation phase (2h10 to 4h50) gave stable activation and memory, even after the removal of the heat shock stimulus (Figure 3D, early).”

d) In order to keep the model and parameters as simple as possible, we have re - run all simulations in Figures 2 and S2 (memory of silencing) with $C_{i,m} = 4$ instead of 8 as in the original version. $C_{i,m} = 4$ was the minimum coupling strength in the original paper that gave good memory of silencing but we originally chose $C_{i,m} = 8$ because it gave more complete suppression of the reporter in the anterior compartment. However in the absence of quantitative experimental information on gene expression levels of the reporter in the original publication, we reason that the slightly less repressed version we obtain with $C_{i,m} = 4$ is sufficiently repressed to make the point that the simple model works.

2.2) The timing of the onset of the histone modifications in Fig 4 do not match the model precisely and some comment could be useful. I suppose there could be detection issues, but we also are not sure exactly how PRE/TRE memory is encoded.

Response: We agree that we do not know how memory is encoded. We have revised the text on page 17, to make clear that there are limits to the extent to which the model predictions can be compared to individual molecular events (as the model considers only A, U and M, which comprise all modifications and bound proteins that contribute to each state). See also response to reviewer 3, point 2.

New text on page 17 of the revised manuscript.

“We note that different histone modifications and PC show different kinetics of detectable accumulation in both the immunofluorescence and ChIP assays, whereas the model predicts a smooth increase of all states during cycles 5-9. In the model, the ‘A’ and ‘M’ system states comprise all modifications and bound proteins that contribute to that state, and thus in its current form the model does not enable predictions about separate molecular events.

The important similarities to the model prediction are that the PC protein and specific histone modifications are present on chromatin at low levels during the early cycles 1-13, accumulate during development, and reach maximum levels during the maintenance phase (stage 10- 14).”

2.3) The match of the experiment and model for the bxd PRE/TRE is not so good in the posterior of the eye disc (compare Fig 5H and P).

Response: We thank the reviewer for pointing this out. This prompted us to revisit the model and the data. Since the model can be simulated an unlimited number of times, the average expression level predicted by the model comes close to a true simulated average. However in the real eye discs, the variegating pattern is highly variable from disc to disc, so that the measured levels may be biased by the stochastic nature of the expression and the small sample size. We had originally performed several evenly spaced line scans on each of several discs, which was sufficient to capture the smooth gradient of the wild type pattern, but was not sufficient for the variegating case. We have now re-analysed the *bxd* disc images by measuring the average intensity profile of the entire disc area. This gives a more accurate measure of the true average expression levels across each disc. The fit between model and data is closer though still not perfect (see new Figure 5H). We cannot determine whether this is due to the large degree of stochastic variation in the data, or a real biological effect that is not captured by the model. We have added a comment on this in the text, page 22.

“We note that in the posterior part of the disc, the model predicted a higher average expression level than was observed in the experiment (compare Figures 5H and P). This may be due to the large variation in degree of variegation in the imaged eye discs.”

In general the paper is very well written and presented, and everything is thoroughly explained. However, I have a few suggestions that would have made it easier for me to follow:

3) In Fig 1C it would aid readers to state “(s=10 sites)” and “(S=40 nucleosomes)” on the figure (or at least in the legend), as these are used throughout, and would save a search in in the Supplementary.

Response: Thanks for this suggestion, we have modified Figure 1C accordingly.

4) In Fig 2 some more explanation and detail of the model could be added by putting a simple table to the right of the 2A panel, showing the change in coupling, p_1 and p_2 (for A and P). Tick marks on the timeline at the left of 2B could be used to illustrate replication events.

Response: Thanks for this suggestion. We have modified Figures 2 and 3 to show the p_1 values used in each time segment and spatial compartment for the promoter simulations, and the C_e and $C_{i,m}$ values for the promoter and PRE/TRE simulations in each time window. We have also added cycle numbers and ticks on the right of Figures 2 and 3 to indicate replication events.

5) In Fig 4D the bars for PC levels could be white to distinguish them from the chromatin bound modifications. Could the apparent lack of PC in the early cycling nuclei be an alternative explanation for the lack of PRE/TRE memory at this stage?

Response: Thanks for this suggestion, we have modified Figure 4D accordingly. Regarding the apparent lack of PC and its effect on memory:

a) Lack of PC.

We think this may be a detection issue. PC protein can be detected in embryos by western blot during the first two hours of development (Steffen et al., NAR 2013, Figure S2 reproduced below). This corresponds to the time window (stage 1-4, cycles 1-13) when it is not visible in nuclei in whole mount staining (current manuscript, Figure 4D and S4). We cannot determine whether PC is on chromatin at these early stages, but undetectable due to low levels. The presence of a low level of PC in the ChIP experiment in this same early time window suggests that at least some of it is on chromatin (Figure 4D).

Figure S2B from Steffen et al., 2013. Staged embryo protein extracts, anti PC western blot. Grey square indicates endogenous PC, black circle indicates a PC-GFP fusion protein. Equivalent amounts of protein were loaded.

- b) Lack of memory. The lack of PRE/TRE memory at these very early stages is a prediction of the model that has not been experimentally tested. The model currently merges all chromatin-associated modifications and proteins that contribute to silencing in the 'M' configuration. So it includes PC, and other chromatin bound PcG proteins, and the histone marks associated with them. Likewise the 'A' configuration merges any chromatin-bound TrxG proteins and their associated histone modifications. It would be very interesting to introduce separate terms in the model for proteins, particularly since we have quantified mitotic binding for some of them (Fonseca et al., 2012, Steffen et al., 2013). Others have observed PcG and TrxG proteins binding replicating chromatin (Francis et al., 2009; Petruk et al., 2012). Such a model would enable the requirement for proteins vs. histone marks to be explored.

We have modified the text on page 6 of the manuscript to make clearer that the model does not distinguish proteins and histone modifications:

"Each of the A and M configurations represents all histone modifications and other bound molecules, such as TrxG and PcG proteins, that contribute to activation or silencing."

On page 24 (Discussion) we have included the following text, prior to the discussion of rapid early cycles:

"In the model, the 'A' and 'M' states each comprise all histone modifications and bound proteins that contribute to that state. We propose that in the living animal, these first 13 cycles may be essential for wiping the system and allowing it to take up new information. This idea is consistent with our experimental observation of low but detectable PC protein and histone modifications in the early cycles, which increase steadily after the onset of cycle 14 (Figure 4, S5). It would be interesting in future to introduce separate terms in the model for PcG and TrxG proteins, some of which have been shown to bind mitotic or replicating chromatin^{46,47}. Such a model would enable the requirement for proteins and histone marks to be separately explored."

Reviewer #2 (Remarks to the Author):

In their article entitled "A theoretical model of Polycomb/Trithorax action unites stable epigenetic memory and dynamic regulation", the authors propose, by quantitative experiments and theoretical modeling, to decipher the role of PRE/TRE genomic modules in the control of gene expression via the PcG/TrxG system. Consistently with a recent computational investigation of the molecular complexity of PcG/TrxG system (Sneppen & Ringrose Nat. Comm. 2019, in press), they introduce a simple effective 3-states model that somehow extends previous related models (in particular the one introduced by Kim Sneppen's and Martin Howard's groups), but that now explicitly introduces a coupling between the promoter (TF-binding) and the PRE/TRE (chromatin/histone marks) states; a main novelty is that now the epigenetic states and the promoter states are considered separately. When confronting the outputs of their stochastic simulations to experimental data (obtained from several published studies or from experimental assays designed and carried out by the authors for the purpose of this article) they show how a minimal version of the model (3 parameters) can recapitulate various experimental observations

associated with different gene regulation logics involving PRE/TRE elements: (I) memory of silencing (II) memory of activation and (III) graded transcriptional response.

A main outcome of this article is the demonstration that the cell cycle duration in the early stages of embryogenesis controls the epigenetic stability of the PRE/TRE states and thus the memory in cis of gene activity: early rapid divisions essentially maintain the PRE/TRE in a naive unmarked state whereas subsequent lengthening of cell cycle drives the system toward bistability (flexibility), where the specific PRE/TRE state is instructed by the corresponding promoter activity and then stably maintained, locking the system in the desired state (fidelity). Such cell cycle duration global constraint was already proposed in previously published pure theoretical studies (Dodd 2007, Zerihun 2016), where it was indeed suggested that stability of epigenetic state during these rapid cell divisions could not be achieved for reasonable parameters values. This present study provides the first clear demonstration of that effect.

Overall, this is a very impressive and elegant study, that manages to combine theoretical modeling and quantitative experiments to provide new and important in-depth understanding of the PcG/TrxG system. The article is well written, very clear and documented, easily accessible to a broad audience ie to scientists from the computational and biological community. Their theoretical and experimental works supporting their demonstrations are very solid and fully convinced me. The results reported in that manuscript open new exciting perspectives in the field of quantitative description of PcG/TrxG-mediated gene regulation and more generally in the field of quantitative epigenetics. This study is a perfect example of how a relatively “simple” theoretical model can provide strong in-depth knowledge when combined with quantitative experiments and thus will undoubtedly encourage further similar collaborative works on related topics.

For all these reasons I recommend the publication of this article in Nature Communication.

Response: We thank this reviewer for thorough reading of the manuscript and for excellent constructive suggestions, which we address below.

I have only few minor concerns:

(1) It is not clear to me how they perform the mapping of the time in their simulation: in other words, how do they fix the 12 s per iteration?

Response: We fixed 12s per iteration empirically. This time window can only be fixed in “real” terms in relation to the known duration of cell cycles. We found that 12 sec per iteration gives sufficient time resolution in the early cycles for the system to undergo several iterations per cell cycle (each of which are 10 mins in the model), but is not so finely grained that the entire model takes too long to run the longer simulations (e.g. over several days of developmental time for the larval simulations). A shorter time window per iteration would work equally well, but would be slower to run and would require re-paramaterisation.

We have added a comment to explain this in the text in the Supplementary methods, page 12, lines 12-16.

(2) The experimental evidence of the influence of cell cycle duration in chromatin state establishment and maintenance has been also obtained by O’Farell’s group in the context of constitutive heterochromatin in *Drosophila* (Shermoen et al., *Curr Biol* (2010), Farrel et al., *Annu Rev Genet* (2014), Yuan et al., *Gene Dev* (2016), Seller et al., *Genes Dev.* (2019) ...). I thus suggest to refer to that work since it relies on a very similar concept.

Response: We thank the reviewer for drawing our attention to this body of very relevant and beautiful work, which we had not previously been aware of. We have cited the review (Farrel, 2014) in the general discussion about early cell cycle length (page 24, new ref. 50).

We have included citations on page 15:

“Thus a prediction of the model is that relevant histone modifications and chromatin bound proteins will be present at low levels from the onset of development and will accumulate slowly during development, reaching maximum levels only during late embryonic development. This is consistent with the observation that heterochromatin features (HP1 and H3K9 methylation) accumulate slowly during cycles 11-13 and increase substantially during cycle 14^{34,60}

In addition we have included the following text on page 25 (discussion):

“Interestingly, a similar effect has been observed for heterochromatin in Drosophila. HP1 and H3K9 methylation progressively accumulate with the progressive lengthening of S - phase during cycles 11-13^{34,60}. This accumulation is increased upon arrest of cycle 13⁶⁰, and abrogated in grp mutants, which have shortened cycles 11-13⁶¹.”

(3)Typos:

- p7, l18; p14, l16; p22 l 18; p28, l6; : extra parenthesis
- p14, l8: “only Supplementaryt to ...” ????
- p15, l12: comma after the reference 36
- p19,l21: missing parenthesis
- p22,17: missing comma after ref 37
- p23, l24: extra point after ref 58

Response: We thank the reviewer for pointing out these typos, which have been corrected in the revised manuscript.

Reviewer #3 (Remarks to the Author):

The paper by Reining and co-authors reports on development of numerical model of Polycomb/Trithorax action that re-capitulates the ability of *Drosophila* PRE/TREs to install epigenetic memory of both repressed and active transcriptional states. Numerical modelling of epigenetic processes is a hot research field that should be of interest to many readers of Nature Communication and the findings of Reining et al. provide substantial advances.

The authors start by formulating the model which behaviour recapitulates classic *Drosophila* experiment. In this experiment the lacZ expression is driven by enhancers of the Ubx gene. During early stages of embryonic development spatial pattern of lacZ expression is defined by interplay of maternally provided repressor and activator proteins that compete for binding to Ubx enhancers. At later stages of development the lacZ reporter becomes active throughout the embryo and larva unless coupled to a PRE/TRE. The authors explore the range of parameters where expression pattern of the reporter simulated by the model match the experiment. They demonstrate that such values exist thereby showing that a simple model can recapitulate some basic features of epigenetic repression. They also show that rapid cell divisions in the early fly embryo and, perhaps as a direct consequence of this, the lack of mutual influence of PRE/TRE state and transcriptional status of the target gene is, in principle, sufficient to explain the “switch-like” behaviour of Polycomb/Trithorax system in the context of *Drosophila* homeotic genes.

Response: We thank this reviewer for thorough reading of the manuscript and for raising several very relevant points, which we address below.

Reviewer's specific comments/ requests (our numbering added).

1) Reining and co-authors then go on to investigate whether the same numerical model run with the same set of parameters can recapitulate epigenetic memory of activation. Here the authors attempt to simulate experiments by Cavalli and Paro. In these experiments the lacZ and white reporter genes are activated by short pulse of GAL4. Given early during embryonic development, GAL4 activates the reporters, which then remain active throughout development. In contrast, if GAL4 is given at later developmental stages, the reporters get transiently activated but then get re-repressed again. The authors claim their model recapitulates results of Cavalli and Paro.

This, at first glance, seems like an overstatement. Yes, the author's model results in the stable expression of the reporters only when the activator is given during early embryonic development. However, in the model, when the activator is given at later stages the reporter is simply not switched on (Figure 3C), which automatically explains the lack of persistent activity. This is different from the results reported by Cavalli and Paro, why made a point of transient activation at later stages without triggering epigenetic memory of active state. Is the model of Reining et al. wrong or the results of Cavalli and Paro a special case? To bolster the support for the model, the authors may want to look at experiments by Poux et al. PMID: 11973279 whose results are, actually, matching those returned by the author's model.

Response: We thank the reviewer for drawing our attention to this point, which was also noted by reviewer 1. We agree this is an important issue and have explored the model further, showing that transient activation is in fact possible with slightly reduced coupling strength between the PRE/TRE and the promoter throughout the simulation. It was not necessary to make the model more complex, simply to relax the coupling ($C = 4$ instead of $C = 8$). We have made the following changes to document this:

- a) We provide a new version of Figure 3, in which all simulations were performed with $C_{i,m} = 4$ (model 1). In the original Figure 3, $C_{i,m} = 8$. This loosening of coupling results in a transient activation of the reporter during the heatshock (Figure 3D, cycle 15). The other conclusions of the model do not change with this lower coupling strength.
- b) Figure S3 now shows models 1 and 2 for $C_{i,m} = 4$.
- c) We have modified the text on page 14 to comment on this:

“In particular, a heat shock given after the end of the initiation phase (4h50) allowed transient activation of the reporter but failed to induce stable switching and memory (Figure 3D, late). In contrast, a heat shock given during the initiation phase (2h10 to 4h50) gave stable activation and memory, even after the removal of the heat shock stimulus (Figure 3D, early).”

- d) In order to keep the model and parameters as simple as possible, we have re - run all simulations in Figures 2 and S2 (memory of silencing) with $C_{i,m} = 4$ instead of 8 as in the original version. $C_{i,m} = 4$ was the minimum coupling strength in the original paper that gave good memory of silencing but we originally chose $C_{i,m} = 8$ because it gave more complete suppression of the reporter in the anterior compartment. However in the absence of quantitative information on gene expression levels of the reporter in the original experiments, we reason that the slightly less repressed version we obtain with $C_{i,m} = 4$ is sufficiently repressed to make the point that the simple model works.
- e) Regarding the discrepancy between the results of Cavalli and Paro and those of Poux et al., we thank the reviewer for drawing our attention to this. The fact that the model can actually recapitulate both situations (transient activation and no activation) is interesting, and raises the possibility that the differences observed experimentally are not fundamental differences of mechanism but rather may be due to small quantitative differences in coupling or strength of activation in the two experiments. We have included a citation of Poux et al and a comment to this effect on page 14 of the revised manuscript.

“Interestingly, in a similar experiment, other authors did not observe transient activation with a late heat shock, but rather continuous repression of the reporter in the presence of a PRE/TRE {Poux, 2002 #42}. We found that the model is also able to recapitulate this lack of transient activation if coupling strength is doubled ($C_{i,m} = 8$; data not shown). This suggests that small changes in quantitative parameters rather than fundamental mechanistic differences may explain different experimental outcomes, and underlines the potential of the model for exploring apparently conflicting results.”

2) The underlying assumption of the model is that the PRE/TRE- mediated memory of transcriptional state is based on modified histones. This is well demonstrated for Polycomb repression, less so for Trithorax system. The model requires that in the early embryos histone modifications at PRE/TRE remain low.

Response: The ‘A’ and ‘M’ states in the model comprise all histone modifications and chromatin bound molecules that contribute to the active and silent configurations respectively, and does not assume that histones alone carry memory. We have modified the text on page 6 to make this clearer (see also supplementary methods, in which the model assumptions are discussed in detail.)

Page 6:

“Each of the A and M configurations represents all histone modifications and other bound molecules, such as TrxG and PcG proteins or non coding RNAs that contribute to activation or silencing. Thus the model makes no assumptions about the molecular nature of memory.”

3) From this the authors “predict” that H3K27me3 (the mark of Polycomb repression) and H3K36me2, H3K36me3, H3K4me1, H3K4me3 (presumed marks of Trithorax system) are low in the early embryos and accumulate slow during development. The authors test their “prediction” by immunostaining of embryos and something they call “ChIP followed by whole-genome qPCR”. In my opinion, this set of experiments is out of place. First, the increase in the bulk levels of H3K27 and H3K36 methylation in the developing fly embryos was already observed by mass-spectrometry (Bonaldi et al., 2004 PMID: 15188406) and, for H3K27 and H3K4, by IF and ChIP-seq (Zenk et al., PMID: 28706074).

Response: We appreciate the reviewer’s concern that other authors have examined accumulation of histone modifications during embryonic development. We were careful to draw the distinction between these studies and our work in the original manuscript (Li et al., 2014, Zenk et al 2017). We thank the reviewer for drawing our attention to the mass spectrometry study, which we had not taken into consideration, and which we have now cited in addition. Our analysis is important for the current study because it extends the time window over which the accumulation of modifications and bound proteins is observed, beyond stage 5, at which previous studies stopped. We have modified the text on page 15-16 in the revised manuscript to make this clearer:

“The prediction that histone modifications are present during cycles 1-13, and increase later, is consistent with mass spectrometry analysis {Bonaldi, 2004 #781}, and with two studies that have examined histone modifications using chromatin immunoprecipitation (ChIP)^{34,35}. In {Bonaldi, 2004 #781}, embryos of 0-3h, 3-6h and 6-9h were examined, thus the transition to cycle 14 at 2h10 was not distinguished. In contrast, both of the ChIP studies^{34,35} addressed this transition, detecting low levels of histone modifications during cycles 1-13, and an increase in global levels during the first 40 minutes of cycle 14 (stage 5, 2h10 – 2h 50). However, neither of these studies extended the analysis beyond stage 5, thus they do not enable comparison to the model prediction that modifications and bound proteins continue to increase beyond stage 5, reaching maximum levels only during the maintenance phase (stages 10-14, 4h20 – 11h20).”

4) Second, the bulk levels of H3K27me3 are done largely by hit-and-run methylation by non-tethered PRC2 (Lee et al., PMID: 25986499; Ferrari et al., PMID: 24289921). This process is not mechanistically related to H3K27 methylation around PREs and, most likely, has different kinetics.

Response: We thank the reviewer for raising this issue, which we agree is important. The parameter p5 in the model covers recruitment-independent conversions. Our aim in examining the developmental accumulation of modifications in the model and in our own and other's experiments is to address the effect of changes in cell cycle length on the global accumulation of modifications and bound proteins, rather than the mechanism by which individual modifications are deposited. We have revised the text on page 17, to make clear that there are limits to the extent to which the model predictions can be compared to individual molecular events (as the model considers only A, U and M, which comprise all modifications and bound proteins that contribute to each state). See also response to reviewer 1, point 2.2.

New text on page 17 of the revised manuscript.

“We note that different histone modifications and PC show different kinetics of detectable accumulation in both the immunofluorescence and ChIP assays, whereas the model predicts a smooth increase of all states during cycles 5-9. In the model, the ‘A’ and ‘M’ system states comprise all modifications and bound proteins that contribute to that state, and thus in its current form the model does not enable predictions about separate molecular events.”

The important similarities to the model prediction are that the PC protein and specific histone modifications are present on chromatin at low levels during the early cycles 1-13, accumulate during development, and reach maximum levels during the maintenance phase (stage 10- 14).”

5) Finally, I do not understand “ChIP followed by whole-genome qPCR” experiments presented on Figure 4E. Do the authors just take all DNA ChIPed with an antibody of interest, amplify it using random primers or adapters and then measure the concentration of amplification products? In this case, why not directly measure the concentration of ChIPed DNA using Q-bit or similar sensitive method and avoid potential amplification biases. In fact, I am puzzled by values shown on Figure 4E. Some of them differ by almost an order of magnitude from those reported by Feller et al., PMID: 25578876 for cultured drosophila cells using well controlled mass-spectrometry.

Response:

- a) The procedure is described in the methods section (see page 32). We do not amplify ChIP material prior to qPCR quantification, but ligate adapters and use these directly as templates for qPCR. Since qPCR enables relative quantification over a large dynamic range we found it to be suitable for this experiment.
- b) Regarding the comparison to mass spectrometry data, we agree that ChIP and mass spec give different numbers. Our aim with the ChIP analysis is to evaluate changes over developmental time for a given modification. Although we normalise each data set to the total amount of histone H3 at the same stage, we do not claim any absolute quantification (as the amounts precipitated will depend on the different antibodies), nor do we intend to draw quantitative comparisons between different modifications. Rather we wish to address how each modification changes over time. Thus we do not expect quantitative similarity to mass spectrometry data generated from S2 cells, in which there are no developmental processes.

6) The paper concludes with an attempt to computationally and experimentally model the developmental regulation of the *eya* gene. The authors claim that at this gene the

Polycomb/Trithorax system does not simply maintain the repressed or active transcriptional state but sharpens the gradient of *eya* expression in the eye-antenna imaginal disc. This part I have the greatest difficulty with.

The authors say that *eya* gene has an enhancer upstream of the transcription start site (TSS) and a PRE/TRE in the first intron. Therefore they make a reporter construct where they fuse the upstream and 5' parts of the *eya* (fragments A and B) to the GFP open reading frame (Figures 5C, S6). Then they look at GFP expression from this construct as well as its truncated variants lacking the enhancer or the part of the first intron that contains presumptive PRE/TRE. They further compare the GFP expression from these constructs to the GFP expression from the matching set of constructs where the "PRE/TRE" part of the first intron is replaced with *bxd*-PRE of the *Ubx* gene. While the construct with *bxd*-PRE showed stochastic loss of *eya* expression in the eye-antenna imaginal disc (Figure 5G), the original construct gave GFP expression gradient resembling that of endogenous *eya* expression (Figure 5C).

The authors suggest that the two PRE/TREs are inherently different and try to model this. Here is where I get lost. I do not understand why the authors say that *eya* PRE/TRE is in the first intron. From genome-wide mapping (attached is the IGB screen shot with ChIP-seq data from Kahn et al., PMID: 27557709, but modENCODE data shows the same) it appears that there is a major PRC1 and PRC2 binding site just downstream of the *eya* enhancer and just upstream of its TSS. From all we know, that is the major PRE/TRE, which is included in all author's constructs. Even if the first intron contains another weaker PRE/TRE, replacing that region with strong *bxd*-PRE results in constructs with two major PREs instead of the one. This provides the simplest explanation of why the construct with *bxd*-PRE shows stochastic loss of eye expression while the original construct does not.

Response: We thank the reviewer for raising this issue. We realise that we did not make it clear in the original manuscript why we chose to work on the intronic PRE/TRE. We apologise for the confusion. The reviewer is correct that there is a ChIP peak for PcG proteins between the enhancer and the TSS. We now show ChIP profiles (drawn from ModEncode data) for PSC, GAF and H3K27me3 for this region and the intronic region (Figure S6A). The reviewer is also correct that other PRC1 and PRC2 proteins show enrichments on the promoter site, which we mention in the legend to Figure S6. The promoter site is also enriched for GAF and H3K27me3. The intronic site shows enrichments for GAF and H3K27me3. We focused on this intronic site for four reasons:

- a) The promoter site covers the TSS of the *eya* gene, and any perturbation by deletion would also disrupt the gene promoter, which we need for our reporter gene activity.
- b) The promoter site contains fewer sequence motifs that are typical of PRE/TREs than the intronic site. The region of highest PcG and GAF enrichment highlighted in Figure 6A contains two PHO binding motifs (GCCAT) spaced 400 bp apart, whereas the intronic site contains five PHO motifs that are closely clustered (within 300 bp) and three more flanking this region (see new Figure S6A). Curiously, the promoter site, which is more highly enriched than the intronic site for the GAF protein in the ChIP experiment, contains only a single GAF binding motif (GAGAG), whereas the intronic site contains a cluster of five consecutive GAF binding motifs (documented in the new version of Figure S6). Based on the clustering of PHO and GAF motifs, we chose to work on the intronic sequence.
- c) Presence or absence of ChIP enrichments is not sufficient to define presence or absence of PRE/TRE function, or to infer PRE/TRE "strength". ChIP enrichments may arise from looping, spreading or as "Phantom peaks" (Jain et al., NAR 2015, PMID 26117547). Lack of ChIP enrichment in a given tissue does not mean that the element is not a PRE/TRE in another tissue. Functional reporter assays are required to define a given candidate sequence as a PRE/TRE. A 3kb region containing the intronic *eya* PRE/TRE and not the promoter site has been shown to function as a bona fide PRE/TRE in a reporter assay in which it was tested without the rest of the *eya* locus (repression of *mw* reporter and genetic dependence on PcG and TrxG, Ringrose et al., Dev. Cell 2003, PMID 14602076). The promoter site has not been tested in such assays.
- d) We present new experimental evidence that the intronic but not the promoter site is required for PcG -mediated regulation in the context of the *eya1::GFP* construct (see response to point 6 below).

We added a comment on page 18 of the revised manuscript:

*“The *eya* 5’ region contains two potential PRE/TREs that are enriched for PcG and/or TrxG proteins in ChIP seq data sets (Figure S6A). One putative PRE/TRE is at the promoter and a second is in the first intron. The intronic PRE/TRE is well-characterised in reporter assays and contains a high density of binding sites for the PHO, GAF and Zeste proteins^{14,17} (Figure S6A). We chose to analyse the function of the intronic PRE/TRE in this study, as the putative promoter PRE/TRE spans the transcription start site, which is required for reporter expression.”*

6) Note, although the “PRE/TRE” region of the first intron seems necessary to make sharp *eya* gradient, there is no evidence that its action depends on Polycomb or Trithorax proteins. Unless I overlooked something, I am afraid the authors have misunderstood the organization of *eya* gene and designed their experiments in a way that their results are hard to interpret.

Response: We thank the reviewer for this observation. We have now added additional data to Figure S6, giving two lines of evidence that the intronic *eya* PRE/TRE is regulated by Polycomb proteins.

- a) We show that this region is required for genetic interaction with PcG proteins. A reporter in which the intronic PRE/TRE is deleted, but that still contains the promoter PRE/TRE, shows no derepression in a ph410 background. In contrast, a reporter in which the intronic PRE/TRE is present, shows derepression in a ph410 background.
- b) We show that mutation of the eight PHO recognition motifs in this intronic PRE leads to a strong derepression of the *mw* reporter, to a similar extent to the derepression observed upon deletion of the entire 1.6 kb. (Figure S6S, T). We note that the putative PRE/TRE at the promoter site is still present in this construct and in the original *eya1::GFP ΔPRE/TRE* construct, but fails to repress the *mw* reporter. This strongly suggests that the repression mediated by the intronic PRE/TRE is (i) stronger than that mediated by the promoter PRE/TRE, and (ii) that the function of this intronic PRE/TRE depends on the PcG protein PHO.

We summarise the above evidence in the revised manuscript on page 19-20:

*“Deletion and mutation analysis of the *eya1::GFP* transgene confirmed that the intronic PRE/TRE is required for PcG - dependent repression of the *mw* reporter. In the absence of this intronic PRE/TRE (Figure S6D, L), or in a version in which binding sites for the PcG protein PHO are mutated (Figure S6S, T), the *mw* reporter was strongly derepressed. Although the putative promoter PRE/TRE was present in these constructs, it failed to repress the reporter. Furthermore, constructs including the promoter PRE/TRE but lacking the intronic PRE/TRE failed to respond to mutation of the PcG gene *polymoeotic* (*ph*) (Figure S6F, R) whereas those containing the intronic PRE/TRE showed derepression in a *ph* mutant background (Figure S6E, Q). Thus we reason that the intronic PRE/TRE is necessary for PcG mediated *eya* regulation, whilst the promoter site alone is not sufficient.”*

7) To summarize, I suggest that the authors do some editing of the “The same model recapitulates epigenetic memory of activation”, remove the “Chromatin modifications increase from low to high levels during *Drosophila* development as predicted by the model” and “Two PRE/TREs respond differently to a dynamically regulated promoter” parts and revise the discussion accordingly. This being done, I recommend the paper to be published by Nature Communication.

Response: We have made changes to “memory of activation” according to the reviewer’s requests, and have clarified our reasoning in the section on chromatin modifications. We have added new data reinforcing the role of Polycomb group proteins in regulating the intronic *eya* PRE/TRE. We feel this has led to a much - improved manuscript and we thank the reviewer for constructive criticisms which contributed to these improvements.

Reviewers' comments:

Reviewer #1 (Remarks to the Author):

I am happy with the responses and modifications with regard to my comments and also those of the other reviewers.

Reviewer #2 (Remarks to the Author):

I consider that the authors have properly addressed the few points that I raised as well as all the different points raised by Reviewer 1 and 3. As such, I strongly support the publication of this revised manuscript.

Reviewer #3 (Remarks to the Author):

The authors made multiple improvements to the original manuscript which addressed my concerns regarding the first part of the work e.g. modelling epigenetic memory of silencing and activation.

The revision and rebuttal did not alleviate my concerns regarding the second part.

As I already commented, the author's model predicts chromatin changes around PRE/TREs. But the bulk of histone modifications happen in parts of the genome that have no PRE/TREs. So there is no reason to expect that the bulk and the specific changes around PRE/TREs are correlated. Hence, there is no reason to expect that the model predicts what should happen to the overall histone pool. I was concerned that the authors used uncalibrated and unverified immunoprecipitation method to measure changes in the abundance of histone modifications during development and still got "experimental proof" of their computational predictions. The authors can now consult the work from Vermeulen's, Imhof's and Muller's labs (<https://doi.org/10.1016/j.devcel.2019.09.011>), which includes 2-4h and 14-16h developmental time points, to see for themselves that their measurements are incorrect. I once again advise the authors to avoid embarrassing themselves and exclude the part "Chromatin modifications increase during Drosophila development as predicted by the model" from publication.

The story with modelling of the eye locus has also not been resolved. It is very good that the authors agree that there is second stronger ChIP signal for Polycomb proteins at the TSS of the eye gene. In the rebuttal the authors write "Presence or absence of ChIP enrichments is not sufficient to define presence or absence of PRE/TRE function, or to infer PRE/TRE "strength". ChIP enrichments may arise from looping, spreading or as "Phantom peaks" (Jain et al., NAR 2015, PMID 26117547)." This is all fine, but I yet to see an example of a strong ChIP signal detected with antibodies against multiple Polycomb proteins that does not behave as a PRE. That is whose DNA does not generate new PcG/Trx binding site when integrated elsewhere in the genome. If the authors are serious about their claim that the TSS-proximal binding site is an artefact and cannot, by itself, recruit Polycomb complexes, they need to prove this. For example, by doing ChIP-qPCR over the TSS in their *eya1::GFP ΔenΔPRE* transgene. Until then, we have to assume that the transgenes described by the authors (original full length and *eya1::GFP ΔPRE+bx* PRE/TRE) contain two PREs not one. This is different from what was simulated in their computational model and, hence, the model and the experiment do not match.

I appreciate the author's efforts to resolve the issue by genetic experiments and that they report all phenotypes although not all of them add up in a simple narrative. In the rebuttal the authors write: "We show that this region is required for genetic interaction with PcG proteins. A reporter in which the intronic PRE/TRE is deleted, but that still contains the promoter PRE/TRE, shows no derepression in a ph410 background. In contrast, a reporter in which the intronic PRE/TRE is

present, shows derepression in a ph410 background.” This is not the full story. Indeed, the reporter that lacks eye enhancer but has the promoter and intronic PRE/TREs present shows more white gene expression on heterozygous ph[410] background (Figure S6Q), while the reporter that lacks both eye enhancer and intronic PRE/TRE but has the promoter PRE present does not show change in the white gene expression on heterozygous ph[410] background (Figure S6R). What the authors overlooked, is that the reporter that contains eye enhancer, promoter PRE/TRE and intronic PRE/TRE also does not change white expression on the heterozygous ph[410] background (Figure S6O). This argues that the change in white expression on ph[410] background is not linked to presence or absence of intronic PRE/TRE but depends on something else. For all we know, in the case where ph[410] mutation increases white expression, the effect may still be mediated by the promoter PRE/TRE. It does not help that the ph[410] mutation chosen for the experiments is hypomorphic. It disrupts ph-proximal (ph-p) gene but leaves closely related paralogue ph-distal (ph-d) intact. ph[410] is known to have variable effects on different PRE containing transgenes for reasons that are not understood (see: Hodgson, J.W., Cheng, N.N., Sinclair, D.A.R., Kyba, M., Randsholt, N.B., Brock, H.W. (1997). The polyhomeotic locus of *Drosophila melanogaster* is transcriptionally and post-transcriptionally regulated during embryogenesis. *Mech. Dev.* 66(1-2): 69--81.).

The second line of author's argument has its own problems. The authors write: “We note that the putative PRE/TRE at the promoter site is still present in this (*eya1::GFP ΔPHO*) construct and in the original *eya1::GFP ΔPRE/TRE* construct, but fails to repress the mw reporter. This strongly suggests that the repression mediated by the intronic PRE/TRE is (i) stronger than that mediated by the promoter PRE/TRE,...”. That is not correct. What this result shows is that, when the eye enhancer is present, the promoter PRE/TRE is not sufficient to repress white expression and that the intronic region deleted in 1.6kb deletion (what the author call Δ PRE/TRE) is required for the repression. The contribution of the intronic region could be very small but enough to tip the balance towards repression and the repression may always require promoter PRE/TRE.

Less important for the argument are experiments with transgenes that have putative intronic Pho binding sites mutated. Here the authors say: “We show that mutation of the eight PHO recognition motifs in this intronic PRE leads to a strong derepression of the mw reporter, to a similar extent to the derepression observed upon deletion of the entire 1.6 kb. (Figure S6S, T).” Just for the record, that is not what I see on the pictures. When I compare Figures S6S, T to Figures S6K, L, I see that the eyes of *eya1::GFP ΔPRE/TRE* (1.6kb intronic PRE deletion) flies are obviously more red than the eyes of *eya1::GFP ΔPHO* flies (transgene with mutated Pho sites). This may mean that mutation of Pho sites does not completely abolish the binding of Polycomb complexes or that 1.6kb deletion has additional stimulatory effects on white expression. For example, by removing a Polycomb-independent repressor element or by bringing eye enhancer closer to white promoter. We also need to remember that the authors provide no direct experimental evidence that the intronic site binds Pho and that this binding is abolished by mutations in Δ PHO transgene.

To sum up, the *eya* locus is more complex than the authors imagine and more experiments have to be done to characterize all its regulatory elements before one can model it in meaningful way. Until then, any resemblance of the author's computational modelling and transgenic experiments may well be explained by chance. I reiterate my advice to drop the dubious *eya* modelling part and publish smaller coherent paper where all results are bomb proof instead of a bigger paper, parts of which are likely to be wrong. If the authors choose to keep the eye modelling part, I request that they move panel A from Figure S6 (ChIP screen shot) to the main figure and amend the schematics of transgenes on Figure 5C and 5G such that the reader can easily see that there is a strong Polycomb ChIP-seq peak at *eya* promoter and that this part is included in all transgenic constructs.

Reviewer 3 comments.

1) a) As I already commented, the author's model predicts chromatin changes around PRE/TREs. But the bulk of histone modifications happen in parts of the genome that have no PRE/TREs. So there is no reason to expect that the bulk and the specific changes around PRE/TREs are correlated. Hence, there is no reason to expect that the model predicts what should happen to the overall histone pool.

Response:

We thank the reviewer for this comment, which prompted us to clarify the explanation of what the model does, both for the reviewer, and in the text of the manuscript.

The model comprises an array of nucleosomes and a promoter. In the cases in which we use the model to investigate promoter- PRE/TRE interactions for specific genes, the nucleosomal array is coupled to the promoter, which is adjusted in the model to reflect its experimentally observed regulation. In these cases, the recruitment of A or M modifications to the PRE/TRE is determined by three processes in the model:

- 1) The activity of the promoter, which, when coupled to the PRE/TRE, dictates whether the PRE/TRE will predominantly accumulate active or silent modifications in the model.
- 2) The strength of feedback (governed by the parameters p3 and p4). This determines the rate at which an existing modification (A or M) will be copied within the array to make a new one of the same kind.
- 3) The rate of feedback-independent transitions (governed by the parameter p5). This determines the rate at which any nucleosome can acquire or lose an A or M modification, independently of a pre- existing modification of the same or opposite kind.

In the example shown in Figure 4, we used the same model but with an important difference, namely the nucleosomal array was not coupled to a promoter. In this way we sought to examine the behaviour of the system in the absence of promoter- induced recruitment, and to examine how it responds to cell cycle length. Indeed this version of the model intentionally ignores specific PRE/TRE behaviour, and instead addresses global, non- recruited nucleosomal dynamics. This is essentially what has been done in many previous publications and is not new (E.g., Dodd et al., 2007 ref. 21; Zerihun et al., 2015, ref. 31). Previous publications have shown that the stability of such a system (i.e. one that is not coupled to a promoter and can adopt A, U, or M states in a bistable manner) is reduced when cell cycles are shorter, and increased when cell cycles are longer. We acknowledge this work clearly in the manuscript (pages 13 and 21). However these publications did not apply these observations to a real developmental scenario, where cell cycle length changes over development. Our purpose here is not to predict an absolute level of "A" or "M" modifications, but rather to examine the expected behaviour of the system over time under the very interesting specific changes in cell cycle length that occur during *Drosophila* embryogenesis, and to explore the potential contribution of the early exit of pole cells to the system behaviour.

We have modified the text on page 13 (lines 9-18) of the manuscript to clarify what we are doing:

" Several previous theoretical studies of chromatin based epigenetic memory have shown that stability of memory in bistable model systems increases with increasing cell cycle length^(21,31). However these models have not been applied to the observed real changes in cell

cycle length that occur during development. To examine the effect of developmental changes in Drosophila division cycle length on the rate of global accumulation of A and M states in the model, we ran simulations over a developmental time course. We ran 1000 simulations on a nucleosomal array that was not coupled to a promoter, over 11 h of development and scored the average levels of A and M states at different time points (Fig 4A). This showed that in the model, A and M states are present at low levels from the onset of development and accumulate slowly during development, reaching maximum levels only during late embryonic development..”

b) I was concerned that the authors used uncalibrated and unverified immunoprecipitation method to measure changes in the abundance of histone modifications during development and still got “experimental proof” of their computational predictions. The authors can now consult the work from Vermeulen’s, Imhof’s and Muller’s labs (<https://doi.org/10.1016/j.devcel.2019.09.011>), which includes 2-4h and 14-16h developmental time points, to see for themselves that their measurements are incorrect. I once again advise the authors to avoid embarrassing themselves and exclude the part “Chromatin modifications increase during Drosophila development as predicted by the model” from publication.

Response:

We thank the reviewer for drawing our attention to the work of the Vermeulen, Imhof and Müller labs and to several deficiencies in our original description of the ChIP method we have used.

We address separate points below.

i) The reviewer states that our ChIP method is uncalibrated and unverified. We apologise for having explained the method too briefly in the original methods section, based on the erroneous assumption that the method would be familiar to readers. We realise that the method of linker ligation to ChIP products followed by PCR amplification using primers that bind to the linkers is documented in an early publication with which readers may not be familiar (Orlando et al., 1998, ref 61). Many publications in the pre ChIP-seq era used this method but we realise it is no longer standard practice. However this does not mean that its usefulness is at an end. We have added a detailed description of the entire procedure and the measures we have taken to control quality and reproducibility in the methods section (pages 27 to 30, new Supplementary Figure 6F). The new step we have added is to use the linker-ligated material directly in qPCR to evaluate ChIP enrichments for the whole genome. We found this to be highly sensitive and quantitative over a large dynamic range (allowing detection of lower concentrations than direct measurements by Q-bit). We describe the measures taken to ensure that there is no amplification bias, and show that independent ChIPs performed in independent chromatin preps give highly reproducible results in this assay (new Supplementary Figure 6F). This analysis prompted us to revisit our ChIP data, and to add a new time point for the period of 6-8 hours, which was previously lacking (Figure 4E). We also revised the method by which we calculated enrichments. We previously calculated enrichments relative to H3 ChIP with the following formula: $2^{(Ct_{input} - Ct_{ChIPx})} / (2^{(Ct_{input} - Ct_{H3\ ChIP})})$. However this expression simplifies to $2^{(Ct_{ChIP\ H3} - Ct_{ChIPx})}$. (In other words: $(ChIP/input) / (H3/input) = ChIP/H3$. We have used this simpler calculation to re-analyse the data leading to smaller error bars and minor changes in some cases. The overall conclusions are unchanged and the calculation is explained in the methods section (page 29). We also explain the vertical scale, see point (ii) below. We thank the reviewer for encouraging us to think more carefully about the presentation of this data and the description of this method.

ii) The reviewer states that our measurements are incorrect when compared to those of the mass spec analysis of the Bonnet et al paper. We think that this objection may arise from the fact that we calculated our ChIP enrichments in Figure 4E as “% Histone H3”. In Bonnet et al., histone modifications are also calculated as % histone H3, but these two measurements are completely different and cannot be expected to yield the same numbers. We now clarify this point in the new methods section (page 29 -30), and here for the reviewer.

In our work we performed ChIP with antibodies against unmodified H3 as a control, and against the modification in question, and calculated the ChIP signal for each modification, as a percentage of that for the H3 antibody in each sample, as described in detail in the new methods section. As with all ChIP experiments, the fact that different antibodies have different affinities for their epitopes means that we cannot perform absolute quantification. We do not intend to claim that we have calculated the % of all H3 molecules that contain a given modification, rather we use the change in relative amounts to compare one time point to another. In the case of Bonnet et al, the authors do indeed directly calculate the absolute % of H3 molecules that carry each modification, using mass spectrometry, which is independent of antibodies. Thus the fact that the y- axis values reported in Bonnet et al. are different to those we show in Figure 4E it is not an indication that our results are incorrect, rather that we are measuring different things.

We thus do not feel that we risk embarrassing ourselves by presenting this data, however the reviewer’s concerns did highlight the fact that we do risk confusing readers if we use the term “% H3”. For this reason we have changed the axis label in Figure 4E to “enrichment relative to H3 ChIP” and expressed the data as a ratio of ChIP enrichments, rather than a %. Thus a relative enrichment of 1 in the new Figure corresponds to “100% H3” in the old figure, but neither scale makes a claim to absolute quantification. We explain the calculation in the new methods section on page 29.

iii) The reviewer points out that the analysis of Bonnet et al includes 2-4h and 14-16h developmental time points. Our ChIP analysis (Figure 4E) includes 0-2, 2-4h, 4-6h, 6-8h and 8-10h, but does not include the 14-16h time point measured by Bonnet et al. Thus the two data sets overlap at only one time point (2-4h). Nevertheless, both studies agree in that they observe an increase in modifications over developmental time, with the exception of H3K36me2 and me3, for which we observe an increase and Bonnet et al observe a decrease between the 2-4h time point and the 14-16h time point. We note that the work of Li et al., (2014) also reports a steady increase in all modifications, including H3K36me3, at early time points, in a ChIP analysis with higher time resolution. We document these comparisons in the new Supplementary Figure 6A-E and include a citation of Bonnet et al in the revised manuscript on page 13.

We reiterate that the main goal of this analysis was not to attempt absolute quantification, but rather:

- To evaluate whether the levels of histone modifications increase in relative terms during development, as suggested by the immunofluorescence data. We and others show that this is generally the case, but none of the other studies covers the same time window as we do (see new Supplementary Figure 6B).

- To determine whether histone modifications that are not detectable in immunofluorescence during the first two hours of development, are in fact detectable by ChIP. We show that they are detectable in the 0-2h time window. This is in agreement with the data of Li et al (2014), but is not addressed by Bonnet et al (2019) (see new Supplementary Figure 6B).

Thus in summary we wish to keep the data of Figure 4 in the paper, and we hope that the new supplementary Figure 6B-E helps to put it in context of other available data, and to show that the method gives reliable and reproducible results (Supplementary Figure 6F).

2) The story with modelling of the eye locus has also not been resolved. It is very good that the authors agree that there is second stronger ChIP signal for Polycomb proteins at the TSS of the eye gene. In the rebuttal the authors write "Presence or absence of ChIP enrichments is not sufficient to define presence or absence of PRE/TRE function, or to infer PRE/TRE "strength". ChIP enrichments may arise from looping, spreading or as "Phantom peaks" (Jain et al., NAR 2015, PMID 26117547)." This is all fine, but I yet to see an example of a strong ChIP signal detected with antibodies against multiple Polycomb proteins that does not behave as a PRE. That is whose DNA does not generate new PcG/Trx binding site when integrated elsewhere in the genome.

If the authors are serious about their claim that the TSS-proximal binding site is an artefact and cannot, by itself, recruit Polycomb complexes, they need to prove this. For example, by doing ChIP-qPCR over the TSS in their *eya1::GFP ΔenΔPRE* transgene.

Until then, we have to assume that the transgenes described by the authors (original full length and *eya1::GFP ΔPRE+bx1 PRE/TRE*) contain two PREs not one. This is different from what was simulated in their computational model and, hence, the model and the experiment do not match.

Response:

As we understand it, the reviewer's main objection to our interpretation is not that the intronic PRE/TRE is not a PRE/TRE. Rather, the reviewer encourages us not to dismiss the promoter site as a potential PRE/TRE. We fully agree, and as outlined below, we have addressed this possibility by modelling and by revisions to the text.

We agree with the reviewer that we cannot exclude the possibility that the promoter proximal ChIP enrichment site is indeed a PRE/TRE. We would be very interested to pursue the analysis of the *eya* locus further, and agree that the experiment suggested by the reviewer would be interesting. However due to the size of the transgene cassette (it includes several kb of flanking sequence on both sides of the *eya* sequences of interest) we were unfortunately not able to design qPCR primers that would exclusively detect ChIP enrichments at the *eya1::GFP* transgene without also detecting the same sequences that are present at the endogenous *eya* locus. We would need to perform mutations or deletions in the endogenous locus using CRISPR-cas, which would be interesting but we feel that this is beyond the scope and aims of the current study.

We accept the reviewer's point of view, namely that there may indeed be two PRE/TREs in the *eya1::GFP* transgene and we propose to proceed accordingly with revisions that take account of this.

The reviewer points out that the model does not match the experiment, if two PRE/TREs are indeed present. This is interesting and we thank the reviewer for prompting us to think more about the potential alternative biological situations. Since we cannot delete the promoter PRE/TRE without affecting the TSS, we instead used the model to explore these alternatives. We present a new Supplementary Figure 10, in which we compare the model in a scenario in which two PRE/TREs are present, and one of them is replaced by the *bxd* PRE/TRE, to the original simpler version, in which only one PRE/TRE is present, and is replaced.

The comparison of the two-PRE/TRE model and the one-PRE/TRE model is described in detail in the supplementary methods section (page 41- 44), and is mentioned in the main text on page 18 (lines 14-19):

*“We implemented two versions of the model, one in which the model reporter locus contains two PRE/TREs (one at the TSS and one intronic, Figure 5A), and a second in which it contains only one intronic PRE/TRE. Both models reached similar conclusions regarding the properties of the *eya* and *bxd* PRE/TREs (Supplementary Figure 10, supplementary methods). Thus we describe the results of the simpler, one-PRE/TRE model here. The two-PRE/TRE model is described in detail in supplementary methods.”*

The most important conclusion to emerge from this analysis is that if there are indeed two PRE/TREs at the *eya::1 GFP* reporter, then they both need to have the inbuilt flexibility to allow the *eya* gradient across the eye disc to exist. It was not possible to model this gradient in the presence of a strong “*bxd*” type PRE/TRE (i.e. one that gives long term memory). Thus the original central conclusion of our work remains unchanged, namely that different PRE/TREs have different inherent properties, allowing either long-term stable memory (like *bxd*) or dynamic fine tuning (like *eya*).

3) a) I appreciate the author’s efforts to resolve the issue by genetic experiments and that they report all phenotypes although not all of them add up in a simple narrative. In the rebuttal the authors write:

“We show that this region is required for genetic interaction with PcG proteins. A reporter in which the intronic PRE/TRE is deleted, but that still contains the promoter PRE/TRE, shows no derepression in a ph410 background. In contrast, a reporter in which the intronic PRE/TRE is present, shows derepression in a ph410 background.”

This is not the full story. Indeed, the reporter that lacks eye enhancer but has the promoter and intronic PRE/TREs present shows more white gene expression on heterozygous ph[410] background (Figure S6Q), while the reporter that lacks both eye enhancer and intronic PRE/TRE but has the promoter PRE present does not show change in the white gene expression on heterozygous ph[410] background (Figure S6R).

What the authors overlooked, is that the reporter that contains eye enhancer, promoter PRE/TRE and intronic PRE/TRE also does not change white expression on the heterozygous ph[410] background (Figure S6O).

Response:

We thank the reviewer for pointing this out. We think that the lack of activation in Figure S6O (now Figure 5O) is due to already fairly high level of eye pigment in the presence of the enhancer. We have modified the text on page 17 as follows, to indicate this and also to be less negative about the potential promoter PRE/TRE:

*“To evaluate genetic interactions we introduced each variant of the *eya1::GFP* transgene into a heterozygous mutant background for the *PcG* gene polymoeotic (*ph*⁴¹⁰). Constructs containing the enhancer were highly expressed and showed no further activation in a *ph*⁴¹⁰ mutant background (Figure 5 O,P). The construct lacking both the enhancer and the intronic PRE/TRE was expressed at lower levels but also showed no response to the *ph*⁴¹⁰ mutation (Figure 5R). In contrast, the construct lacking the enhancer but containing the intronic PRE/TRE showed derepression in a *ph*⁴¹⁰ mutant background (Figure 5Q). Taken together these results indicate that the intronic PRE/TRE plays a role in *PcG* mediated regulation of the *eya1::GFP* reporter.”*

b) This argues that the change in white expression on *ph*[410] background is not linked to presence or absence of intronic PRE/TRE but depends on something else. For all we know, in the case where *ph*[410] mutation increases white expression, the effect may still be mediated by the promoter PRE/TRE.

Response:

We disfavour this interpretation because the reporter that lacks both eye enhancer and intronic PRE/TRE but has the promoter PRE/TRE present does not show a change in the white gene expression in the heterozygous *ph*[410] background (Figure S6R).

c) It does not help that the *ph*[410] mutation chosen for the experiments is hypomorphic. It disrupts *ph*-proximal (*ph*-p) gene but leaves closely related paralogue *ph*-distal (*ph*-d) intact. *ph*[410] is known to have variable effects on different PRE containing transgenes for reasons that are not understood (see: Hodgson, J.W., Cheng, N.N., Sinclair, D.A.R., Kyba, M., Randsholt, N.B., Brock, H.W. (1997). The polyhomeotic locus of *Drosophila melanogaster* is transcriptionally and post-transcriptionally regulated during embryogenesis. *Mech. Dev.* 66(1-2): 69--81.).

Response:

We thank the reviewer for pointing this out. We are aware of this, as one of us (Ringrose) was a post doc in the laboratory of Jean Maurice Dura during the late 1990's when some of the work on *ph* was being performed. We have made clear in the methods section (page 31) that *ph*[410] is hypomorphic, and included a reference to the flybase entry.

4) The second line of author's argument has its own problems. The authors write: “We note that the putative PRE/TRE at the promoter site is still present in this (*eya1::GFP* ΔPHO) construct and in the original *eya1::GFP* ΔPRE/TRE construct, but fails to repress the *mw* reporter. This strongly suggests that the repression mediated by the intronic PRE/TRE is (i) stronger than that mediated by the promoter PRE/TRE,...”.

That is not correct. What this result shows is that, when the eye enhancer is present, the promoter PRE/TRE is not sufficient to repress white expression and that the intronic region

deleted in 1.6kb deletion (what the author call Δ PRE/TRE) is required for the repression. The contribution of the intronic region could be very small but enough to tip the balance towards repression and the repression may always require promoter PRE/TRE.

Response:

That is a really interesting idea, and we were also intrigued by this possibility. We have modified the text as indicated above under point 3a, to remove the statement that the intronic PRE/TRE is stronger. We show with modelling that the results can also be explained by two PRE/TREs of equal strength, and that our original central conclusion remains unchanged, namely that different PRE/TREs have different inherent properties, allowing either long-term stable memory (like *bxm*) or dynamic fine tuning (like *eya*). (See response to point 2 above, Supplementary Figure 10 and supplementary methods (page 41- 44).

5) Less important for the argument are experiments with transgenes that have putative intronic Pho binding sites mutated. Here the authors say: "We show that mutation of the eight PHO recognition motifs in this intronic PRE leads to a strong derepression of the *mw* reporter, to a similar extent to the derepression observed upon deletion of the entire 1.6 kb. (Figure S6S, T)." Just for the record, that is not what I see on the pictures. When I compare Figures S6S, T to Figures S6K, L, I see that the eyes of *eya1::GFP Δ PRE/TRE* (1.6kb intronic PRE deletion) flies are obviously more red than the eyes of *eya1::GFP Δ PHO* flies (transgene with mutated Pho sites). This may mean that mutation of Pho sites does not completely abolish the binding of Polycomb complexes or that 1.6kb deletion has additional stimulatory effects on white expression. For example, by removing a Polycomb-independent repressor element or by bringing eye enhancer closer to white promoter. We also need to remember that the authors provide no direct experimental evidence that the intronic site binds Pho and that this binding is abolished by mutations in Δ PHO transgene.

We thank the reviewer for raising this. We have modified the text on page 17 to remove the statement that the derepression is similar to that observed upon deletion of the entire PRE/TRE:

*"Deletion and mutation analysis of the *eya1::GFP* transgene confirmed that the intronic PRE/TRE is required for PcG - dependent repression of the *mw* reporter. In the absence of this intronic PRE/TRE (Figure 5D,L), or in a version in which binding sites for the PcG protein PHO were mutated, the *mw* reporter was strongly derepressed (Figure 5S,T, Supplementary Figure 7D)."*

We appreciate the reviewer's concern that the genetic experiments may have several interpretations, and have modified the revised text to keep this point open. However we maintain that the derepression observed upon mutations of PHO sites is a strong indicator that PHO may be involved in the repression at the intronic PRE/TRE.

6)a) To sum up, the *eya* locus is more complex than the authors imagine and more experiments have to be done to characterize all its regulatory elements before one can model it in meaningful way. Until then, any resemblance of the author's computational modelling and transgenic experiments may well be explained by chance.

Response:

We have now thoroughly explored both possible scenarios in the *eya* model (one PRE/TRE or two PRE/TREs). We find that the model can well explain both possibilities, and indeed that a similar but even more limited parameter set is required to explain the results if two PRE/TREs are present (Supplementary Figure 10). The fact that specific, limited and non-overlapping parameter sets are required to fit the model (be it with one or two PRE/TREs) the *eya* and *bxd* data shows that the correspondence between the model and the experiment does not arise “by chance” as the reviewer suggests, but that there is a real difference here, and that the model helps to explore why that might be.

b) I reiterate my advice to drop the dubious *eya* modelling part and publish smaller coherent paper where all results are bomb proof instead of a bigger paper, parts of which are likely to be wrong.

Response:

The reviewer advises to avoid publishing a paper containing the modelling of the *eya* locus because parts of it may be wrong. We wish to distinguish here between data and interpretation: Our data and the technical manner in which we have modelled it are correct. However on the basis of current information we cannot know exactly how many PRE/TREs there are at the *eya* locus, thus any interpretation contains some uncertainty. We now make this clear in the manuscript and we also make clear that we use the modelling to explore avenues that we cannot explore experimentally in the scope of the current work.

In the words of the statistician George Box : “Essentially, all models are wrong, but some are useful”. (Box GEP, Draper NR. 1987. Empirical Model-Building and Response Surfaces. New York, USA: John Wiley and Sons). We maintain that although our model is undoubtedly wrong in some respects (as indeed are all models), the assumptions used are well-justified and explained in the manuscript. This inherent “wrongness” of all simplified models should not be a reason not to publish modelling studies. In general, simplified models are useful tools because they identify unifying principles, and they also highlight quantities or mechanisms that are unknown.

Indeed, we show that our model is useful, because it conceptually unifies many different experimental observations, highlights uncertainties, and makes testable predictions. The main conclusions of the paper, namely that the *eya* and *bxd* PRE/TREs must have different inherent properties to account for their effects in the assay, are in fact strengthened by this analysis. We thank the reviewer for encouraging us to explore this angle.

c) If the authors choose to keep the *eya* modelling part, I request that they move panel A from Figure S6 (ChIP screen shot) to the main figure and amend the schematics of transgenes on Figure 5C and 5G such that the reader can easily see that there is a strong Polycomb ChIP-seq peak at *eya* promoter and that this part is included in all transgenic constructs.

Response:

We thank the reviewer for this suggestion and have moved panels A to T of the old Figure S6, to a new main Figure 5. We feel that this brings the debate about the number of PRE/TREs at the *eya* locus more to the forefront and allow the reader to see all relevant data in the main part of the paper.

In conclusion, we wish to reiterate the conceptual advance of our work:

- 1) We show that a single model is sufficient to recapitulate both long-term stable memory and dynamic regulation. The *eya* experiments and model are essential to this insight. The difference between different regulatory modes of PRE/TREs and their target genes has been recognised and examined in individual experiments on a case-by-case basis. Our model provides the first systematic conceptual framework within which to unify these observations.
- 2) Our model is the first to systematically interrogate experimentally observed interactions between PRE/TREs and promoters during development. Although the *Drosophila* epigenetics community has long been aware that promoters (or enhancers) and PRE/TREs are independent and separable units that interact to give different regulatory outputs, and that this is dependent on developmental context, no previous theoretical study has taken account of this fact.
- 3) By combining a large number of past and present experimental observations with a simple but realistic model, this work has the potential to open a new direction in the Polycomb/Trithorax field.

REVIEWERS' COMMENTS:

Reviewer #3 (Remarks to the Author):

Reviewing previous version of the manuscript by Reining et al. I voiced several specific concerns. Because these were not feasible to address conclusively without much additional work I suggested two possible solutions. Either exclude the parts in question from the manuscript or amend the manuscript in a way that alternative interpretations underlying my concerns are adequately exposed to the reader. The authors chose the second path.

I believe my job as a reviewer ends here. Let the scientific community decide what they take from this work.